# Effect of Graphene Oxide Localization on Morphology Development and Rheological and Mechanical Properties of Poly(lactic acid)/ethylene vinyl Alcohol Copolymer Blend Composites: A Comprehensive Study

**DOI:** 10.3390/polym16081061

**Published:** 2024-04-11

**Authors:** Parsa Dadashi, Suprakas Sinha Ray, Amir Babaei

**Affiliations:** 1Advanced Polymer Materials & Processing Lab, School of Chemical Engineering, College of Engineering, University of Tehran, Tehran 14174-66191, Iran; parsa.dadashi@ut.ac.ir; 2Centre for Nanostructures and Advanced Materials, DSI-CSIR Nanotechnology Innovation Centre, Council for Scientific and Industrial Research, Pretoria 0001, South Africa; 3Department of Chemical Sciences, University of Johannesburg, Droonfontein, Johannesburg 2028, South Africa; 4Department of Polymer Engineering, Faculty of Engineering, Golestan University, Gorgan 15759-49138, Iran

**Keywords:** PLA, EVOH, graphene oxide, rheology, lubrication, localization

## Abstract

This study investigates the rheological, morphological, and mechanical properties of melt-processed polylactide/ethylene vinyl alcohol (70PLA/30EVOH) blend composites containing 0.25, 0.5, and 1 wt.% of graphene oxide (GO) nanoplates. Thermodynamic-based suggested the localization of nanoparticles in EVOH, SEM studies showed that the introduction of GO to the blend increased dispersed droplet size, which was attributed to the localization of GO within EVOH, as confirmed by TEM. The rheology results indicated a decrease in the elasticity for the composite containing 0.25 wt.% of GO compared to the neat blend, which was attributed to the sliding effect of the added GO nanoplatelets. However, samples containing higher amounts of GO nanoplatelets exhibited more excellent elasticity than the neat blend. The increased elasticity was suggestively attributed to the dominance of hydrodynamic interactions, the physical network of added nanoplatelets, and polymer/GO interactions over the sliding role of the GO nanoplatelets at higher loadings. In addition, the effect of the order of mixing was investigated, and the premixing of PLA and GO exhibited a decrease in the droplet radius compared to the neat blend. It was ascribed to the localization of GO nanosheets in the PLA and interface, which was confirmed by rheological results and mechanical assessments.

## 1. Introduction

Polylactide (PLA) is an aliphatic and thermoplastic polyester that has gained significant attention due to its compostability, renewability, transparency, and mechanical properties, as reported in several studies [1,2,3]. However, the material’s inherent brittleness at room temperature, low elongation at break, insufficient thermal stability, and weak barrier properties limit its applications [4]. Various methods have been proposed to address these issues, such as copolymerization strategies, plasticization with a miscible component, blending with an immiscible homopolymer, and/or block copolymerization [5,6,7]. These techniques aim to produce PLA-based materials with a broad range of properties and improved processability.

Ethylene vinyl alcohol (EVOH) is a semi-crystalline polymer with excellent barrier and mechanical properties, making it a suitable candidate for blending with PLA to improve its performance [8,9,10]. Blending PLA with EVOH can improve the PLA’s barrier and mechanical properties, making it a more versatile material for various applications [11,12]. For example, Gui et al. [13] investigated the use of PLA/EVOH blends in packaging applications by studying the degradation behavior, phase morphology development, and final properties of the blends as a function of EVOH content. The authors found that the presence of EVOH promoted the degradation of PLA, which affected the viscosities and morphologies of the blends. The study also found that the addition of EVOH enhanced the crystallinity of PLA and increased the barrier properties to water vapor and oxygen in a linear fashion with increasing EVOH content. However, the authors noted that incorporating EVOH into PLA resulted in only a slight change in the tensile and impact properties of the blends. 

Nonetheless, further enhancements in properties are still desirable for specific applications. One influential approach to achieving superior performance is the incorporation of nanoparticles such as graphene oxide (GO) into polymer blends [14]. Incorporating nanoparticles into polymer blends can further enhance their properties, leading to the development of polymer blend nanocomposites with even greater potential [15,16,17]. 

Recently, GO nanoparticles have emerged as a favorable option for developing polymer nanocomposites with numerous applications, owing to their diverse physical properties, biocompatibility, and barrier properties, making them a promising material for a wide range of packaging applications [18,19,20,21,22,23,24]. For example, Butlhoko et al. [25] investigated the effect of the dispersion of graphite and GO on the thermal, mechanical, and rheological properties of the PLA/poly(ε-caprolactone) (PCL) blend composites. They demonstrated that blend composites filled with GO exhibited remarkable thermal stability at a low GO loading (0.05 wt.%) and a higher degree of crystallinity than the neat blend and graphite-filled composites. They also discussed that the addition of more GO to the composites increased the degree of crystallinity but did not improve the thermal stability significantly due to the aggregation of the GO particles in the blend matrix. Elevating the quantity of GO reduced the tensile properties of the composites, but it enhanced the degree of crystallinity and stimulated the blend’s crystallization process [25]. 

In another study, Salehiyan et al. [26] fabricated composites containing 1 wt.% platelet-like nanoparticles (nanoclay or GO), spherical silica nanoparticles, or carbon nanotubes (CNTs) to investigate the effects of localization of different types of particles on the morphology–rheology relationships of the (80/20) PLA/poly(butylene adipate-co-terephthalate) (PBAT) blends. According to their findings, the composites filled with GO and organically modified nanoclay (Cloisite 30B, C30B) exhibited a weaker linear viscoelastic response. In contrast, in the CNT-filled composite, the morphology was stabilized, and the CNTs firmness led to a plateau behavior similar to that of a solid. The reduced viscoelastic properties of the composites filled with C30B and GO were linked to the speeded-up degradation of these nanoparticles at high temperatures.

In our previous study [27], we investigated the interplay of GO and polyamide 6 (PA6) in PA6/acrylonitrile–butadiene–styrene (ABS) polymer blends by conducting morphological, mechanical, and rheological assessments. It was observed that at 0.5 wt.% of GO, the elasticity was lower than that of the neat polymer blend due to the dominance of the lubricating effect over hydrodynamic interactions and GO-PA6 interactions. It was observed that GO partial localization at the interface in the samples prepared with premixing of GO and ABS resulted in improved interfacial adhesion and mechanical properties. 

As previously mentioned, the GO-containing PLA/EVOH blend composites can be suitable for biocompatible packaging applications. Therefore, finding a more profound understanding in terms of the performance and microstructure of this compound is highly necessary. In this regard, in this study, we aim to fabricate composites with different concentrations of GO and varied mixing sequences to address the dual role of GO, which acts as both a lubricant and an elastic material, and the potential compromise between these two functions. Further, investigating the correlation between rheology, morphology, and mechanical properties can aid in developing these materials for commercial applications.

## 2. Materials and Method

### 2.1. Materials

The extrusion grade PLA type 2003D with D-lactide content of 4.3% was supplied by NatureWorks Co. Ltd. (Minnetonka, MN, USA). It had a molecular weight of 200 kg/mol and a melt flow index (MFI) of 6 g/10 min (at 210 °C, 2.16 kg). The EVOH, with commercial name EVAL, was bought from Kurary America, Inc. (Houston, TX, USA). It had a melt flow index of 1.3 g/10 min (190 °C, 2.16 kg) and a molecular weight of 55 kg/mol. Sigma Aldrich Co. (St. Louis, MI, USA) supplied the graphite flakes. Chem-Lab Inc. (Luxembourg, Belgium) provided sulfuric acid (H_2_SO_4_, 99%), hydrochloric acid (HCl, 37%), phosphoric acid (H_3_PO_4_, 85%), and hydrogen peroxide (H_2_O_2_, 30%). Potassium Permanganate (KMnO_4_) was acquired from Scharlab Co. (Barcelona, Spain).

### 2.2. GO Nanoplatelets Synthesis 

In the initial phase, a mixture was formed by combining 360 mL of H_2_SO_4_ and 40 mL of H_3_PO_4_ with 3 g of graphite nanoflakes. The resulting mixture was stirred for 30 min at a temperature of 35 °C. In the subsequent step, 18 g of KMnO_4_ was gradually added to the prepared mixture and then stirred for 24 h at 50 °C. The mixture was diluted with 700 mL of distilled water in a cooling bath. After 15 min, 30 mL of 25% H_2_O_2_ was added to the mix until the release of oxygen ceased. The mixture was centrifuged at 4500 rpm for 10 min, and the resulting precipitated GO was washed three times with 5% HCl. In the subsequent step, the solution was filtered using a dialysis tube until the pH stabilized at 5. The GO solution was then diluted with distilled water and subjected to sonication at room temperature for 8 h. Finally, the prepared solution was sonicated using a high-power sonicator for 10 min at 1000 W to disperse the GO nanoplatelets. Finally, the freeze-drying process was used to remove the GO nanoplatelets (now abbreviated as GO) from the water. The synthesis of the GO nanoparticles in this study followed the method proposed by Marcano et al. [28].

### 2.3. Processing of PLA/EVOH/GO Composites

The PLA/EVOH neat blend and blend composites containing three different loadings of GO were processed using a German-made internal mixer from the Brabender brand. The mixing process took place at 180 °C and 70 rpm. The objective was to examine the effects of different mixing sequences on rheology, morphology, and mechanical properties in blends containing 0.25% GO-NPs. Three distinct mixing sequences were employed:

M. Initially, PLA and EVOH were simultaneously mixed for 5 min. Afterward, GO was introduced to the PLA/EVOH blend, and the mixing continued for 10 min.

N. A 5-min mixing step involved EVOH and GO. Subsequently, the EVOH/GO masterbatch was blended with PLA for 10 min.

O. PLA and GO were mixed concurrently for 5 min. EVOH was then added to the PLA/GO masterbatch.

The composition and mixing protocol for the prepared samples are provided in Table 1, outlining the specific details.

### 2.4. Characterization Methods

The PerkinElmer FTIR analyzer RX1 model (Thermo Scientific Inc., Waltham, MA, USA) was used to verify the functional groups of GO by conducting measurements in the wavenumber range of 1000–4000 cm^−1^. Moreover, 2 mg of nanoparticles were mixed with 200 mg of KBr powder and then pressed to prepare thin circular discs for FTIR analysis. Before FTIR analysis, mixed powder of graphene oxide (GO) and KBr was dried in a vacuum oven for 24 h at 30 °C to minimize moisture content. To confirm the chemical structures of GO, electronic transitions were obtained using a T90+ UV-visible spectrophotometer manufactured by PG Instruments Ltd. Moreover, 10 mg of GO was dispersed in 100 mL of deionized water. The solution was then sonicated for 30 min. After sonication, the solution was diluted with deionized water to achieve a proper concentration for UV-vis analysis. Both the solvent and solution were scanned from 190 to 600 nm.

The dimensions of GO were measured using a NanoWizard II AFM (JPK Instruments, Bruker Company, Berlin, Germany) in tapping mode. Transmission electron microscopy (TEM) analysis was conducted using a Zeiss EM10C (Carl Zeiss Meditec Company, Jena, Germany) operating at 100 kV on a Holy carbon-coated grid Cu Mesh 300 with EMS grade (Electron Microscopy Sciences (EMS) Company, Hatfield, PA, USA). Before the TEM analysis, the GO suspension was sonicated using a Misonix-S3000 sonicator. For TEM analysis of polymer blend nanocomposite films, samples were cryo-microtomed at −100 °C to maintain their morphology. Cryo-microtoming was performed using a diamond knife, and the samples were cut to a thickness of approximately 2.5 µm with an area of about 4 × 4 µm. The morphology and structure of the PLA/EVOH blend and the PLA/EVOH/GO nanocomposites were studied using a MIRA3 field emission scanning electron microscope (FE-SEM) (TESCAN Company, Brno, Czech Republic) with a 20 kV accelerating voltage. To prepare the samples for analysis, they were cooled in liquid nitrogen for 10 min, broken to reveal their fractured surfaces, and then coated with gold to improve conductivity.

The rheological properties of the samples were measured in the linear viscoelastic region using an USD200 rheometer (Anton Paar Company, Graz, Austria) The measurements were conducted at 180 °C under a nitrogen atmosphere using disks with a diameter of 25 mm and a thickness of 1 mm, prepared by compressing the samples at 180 °C and under 100 bar pressure for 5 min. The SAOS frequency sweep tests were performed within the 0.1–600 rad/s range. The linear viscoelastic region of the samples was identified by conducting strain sweep tests at a strain value of 1%.

The study conducted tensile tests using a SANTAM testing machine (STM-20) (Santam Company, Cape Town, South Africa) with a 5000 N load cell and a 10 mm/min strain rate. The experiment was repeated five times to ensure accuracy, and the study reported the average result.

A film sample was microtomed and then examined using Zeiss EM10C (Carl Zeiss Meditec Company, Jena, Germany) TEM at a magnification of 27.800 KX to investigate the localization of GO in PLA/EVOH/GO blend composite.

## 3. Result and Discussion

### 3.1. Characterization of Synthesized GO 

Figure 1 reports the FTIR and UV-Vis spectra of synthesized GO. The FTIR spectrum in Figure 1a demonstrates specific peaks observed at 1103, 1419, 1639, and 1737 cm^−1^, corresponding to the chemical bonds of C-O-C, C-OH, C = C, and C = O in GO, respectively [29,30,31,32,33]. FTIR of graphite in Figure 1b has been compared to GO to validate the functional groups of GO. Furthermore, the UV-Vis spectrum of GO reported in Figure 1b shows two typical peaks at 236 nm and 297 nm, representing the π-π* transition of C = C bonds and the n-π* transition of C = O bonds, respectively [25,34].

The microstructure of the synthesized GO dispersed well in water was examined using TEM and AFM. The results are presented in Figure 2. The TEM images depicted a thin and flexible 2D layer of GO with visible surface wrinkles. The presence of darker regions in the TEM micrograph (Figure 2a) can be attributed to areas where the thickness of GO is increased due to the presence of wrinkles [27,35].

Furthermore, in Figure 2a,b, we can compare the sizes of graphite and GO. The size of graphite is approximately 250 µm (Figure 2b). However, this size is reduced during the synthesis of GO due to the acidic environment and the specific procedures involved in the synthesis of GO. Furthermore, AFM analysis of the synthesized GO suspended in water (Figure 2c,d) revealed that the thickness of GO was approximately 1.3 nm, while the lateral size measured around 160 nm. In summary, based on the observations from Figure 1 and Figure 2, it can be concluded that the synthesis of GO successfully produced mainly single-layer platelets with an oxidized structure.

### 3.2. GO Localization in PLA/EVOH/GO Composite

The localization of the GO nanoparticles in the PLA/EVOH polymer blend composite was investigated using thermodynamic theory. According to Young’s equation (Equation (1)), the localization of nanoparticles relies on both the interfacial tension between polymers and nanoparticles and the interfacial tension between different polymers. $\gamma_{s1}$, $\gamma_{s2}$, and $\gamma_{12}$ are the PLA/GO interfacial tension, the EVOH/GO interfacial tension, and the PLA/EVOH interfacial tension, respectively. Also, the wetting coefficient ($\omega_{12}$) can be obtained from Equation (1). If the value of ω is greater than 1 or less than −1, it indicates that the nanoparticles tend to localize within the polymer phases. However, when the value of ω falls between −1 and 1, it suggests that the nanoparticles preferentially localize at the interface of polymers [36]. The harmonic equation (Equation (2)) can be employed to calculate the interfacial tension for materials with low energy characteristics, such as polymers. This equation provides a suitable method for quantifying the interfacial tension between these materials. Also, the geometric equation (Equation (3)) calculates the interfacial tension of high-energy materials, such as nanoparticles. This equation offers a reliable approach to quantifying the interfacial tension between these materials. $\gamma_{i}$ represents the surface energy of the “i” sample, while $\gamma_{i}^{d}$ and $\gamma_{i}^{p}$ represent the dispersed and polar components of the surface energy, respectively [37].
(1)$$\omega_{12}=\frac{\gamma_{S2}-\gamma_{S1}}{\gamma_{12}}$$
(2)$$\gamma_{12}=\gamma_{1}+\gamma_{2}-4[\frac{\gamma_{1}^{d}\gamma_{2}^{d}}{\gamma_{1}^{d}+\gamma_{2}^{d}}+\frac{\gamma_{1}^{p}\gamma_{2}^{p}}{\gamma_{1}^{p}+\gamma_{2}^{p}}]$$
(3)$$\gamma_{12}=\gamma_{1}+\gamma_{2}-2(\sqrt{\gamma_{1}^{d}\gamma_{2}^{d}}+\sqrt{\gamma_{1}^{p}\gamma_{2}^{p}})$$

Table 2 displays the surface energies of the PLA, EVOH, and GO materials. The interfacial tensions were calculated using Equations (2) and (3) and are reported in Table 3. Then, ω_12_ was calculated using Equation (1), resulting in a value of 5.3. Therefore, it can be concluded that the GO nanoparticles thermodynamically prefer to localize in the EVOH.

### 3.3. Morphological Evaluation

Figure 3 depicts the SEM micrograph of the cryo-fractured surface of PLA/EVOH and its composites with a composition of 70/30 (*w*/*w*). According to the observation from the SEM results, it is evident that this blend exhibits a dispersed-matrix morphology. In this morphology, the polymer with a higher content (PLA) tends to act as the matrix, while the polymer with a lower content (EVOH) tends to form the dispersed phase. Notably, the cracked plates observed in Figure 3c,d are related to the fractured surface during cryo-fracturing of the sample. The numerical average diameter (R_n_) and volume average diameter of droplets (R_v_) were obtained by Equations (4) and (5), respectively. R_i_ and n_i_ are the radius and number of the droplet of i. R_i_ was obtained from Figure 3 by selecting 100 droplets. Also, the distribution of the size of droplets was obtained by Equation (6). R_n_, R_v_, and distribution of each sample are shown in Table 4.
(4)$$R_{n}=\frac{\sum R_{i}n_{i}}{\sum n_{i}}$$
(5)$$R_{v}=\frac{\sum n_{i}R_{i}^{4}}{\sum n_{i}R_{i}^{3}}$$
(6)$$Dispersity=\frac{R_{v}}{R_{n}}$$

In addition, the SEM micrographs of the PLA/EVOH and nanocomposites containing 0.25 wt% of GO prepared by different mixing protocols were obtained and are presented in Figure 4. Two different mixing protocols were used as follows: in one ((PLA/0.25GO)/EVOH), GO was premixed in the PLA phase, while in the other ((EVOH/0.25GO)/PLA), it was premixed in the EVOH phase. The results indicated that when the nanoparticles were localized in the PLA phase, the EVOH droplet size decreased due to the localization of GO nanoparticles at the interface.

According to Table 4, the Rv values were 1.01 µm, 1.94 µm, and 2.84 µm for PLA/EVOH, (PLA/EVOH)/0.25GO, and (PLA/EVOH)/0.5GO samples, respectively. The R_v_ value of the (PLA/EVOH)/1GO sample was measured to be 2.83 µm, which is roughly similar to the (PLA/EVOH)/0.5GO sample but slightly lower. This trend is also observed in the case of R_n_. Moreover, the enlargement of droplets can be observed by introducing GO to the PLA/EVOH blend. Additionally, the R_v_ increased from 1.94 to 2.84 µm with an increase in the GO content from 0.25 to 0.5 wt.%. However, with a further increase in the GO content to 1 wt.%, the R_v_ decreased to 2.83 µm. 

The observed increase in droplet radius compared to the neat blend can be attributed to the predominant localization of nanoparticles in the EVOH phase, which may have suppressed droplet breakup and accelerated droplet coalescence. The morphological assessments are in harmony with the thermodynamic prediction. 

A TEM analysis was conducted on the (PLA/EVOH)/0.25GO sample to elucidate the specific localization of the GO particles. Figure 5 provides a visual representation of the localization of GO within the EVOH phase when 0.25 wt.% GO was loaded. In Figure 5b,c, dark regions, especially in the top, right, and bottom left of Figure 5c, can be observed. These dark regions may be related to the PLA. The increase in droplet size resulting from the incorporation of a higher amount of GO (0.5 wt.%) into PLA/EVOH can be attributed to the enhanced droplet coalescence, subsequently leading to larger droplets [15,42].

Furthermore, thermodynamic evaluation aligns with microscopy results regarding GO localization. As evident from the TEM results, the thickness of the GO nanoplates in the polymer matrix is approximately greater than 50 nm, and their lateral size is about 1 to 2 µm. This suggests that the GO stacks have more than 50 layers, resulting in an observable lubrication effect in rheological assessments. The stacking process occurs when the GO nanoparticles are introduced to PLA/EVOH because of the stronger interactions between the GO particles compared to the interaction between the polymer and GO. Other TEM and AFM results shown in Figure 2 are related to the synthesized GO not incorporated into the polymer. Blending GO with polymer leads to the agglomeration of particles, increasing thickness and lateral sizes to some extent.

Microstructure analysis brought important results, which can be stated as follows:Droplet radius reduction, which was observed when the GO content increased from 0.5 wt% to 1 wt%, was a consequence of the migration of the GO particles to the PLA and the interface of PLA/EVOH. Increasing the amount of the GO particles can compete kinetically with the thermodynamic affinity of the GO to EVOH droplets, thereby increasing the possibility of localizing GO in other phases (PLA and the interface) besides EVOH. Particle localization in PLA and the interface was an important factor in decreasing the size of the EVOH droplets due to the decrease in the coalescence rate of the EVOH droplets [35].When the GO content was increased from 0.25 wt% to 0.5 wt%, the size of EVOH droplets increased. The localization of GO in PLA and at the interface was insufficient for controlling the coalescence and breakup of droplets compared to the GO distributed in EVOH.According to Table 4, when the GO nanoparticles were premixed with EVOH, the EVOH droplet radius increased from 1.01 to 3.13 µm. This increment is attributed to the predominant localization of GO in EVOH due to premixing with EVOH, which reduced the breakup rate of droplets.When GO was premixed with PLA, the radius decreased to 0.94 µm. The decrease in droplet radius for the sample (PLA/0.25GO)/EVOH could be attributed to the mixing protocol used, where PLA was first mixed with GO before EVOH was added to the PLA/GO blend. The migration of GO from PLA to EVOH during mixing could be explained by the higher affinity of GO to EVOH. Subsequently, the GO particles became trapped at the interface of the PLA/EVOH blend, causing a reduction in the interfacial tension between the two phases. The obtained results demonstrate that the mixing protocol used had a notable impact on the morphology and size of dispersed droplets in blend composites.

### 3.4. Rheological Assessments 

Rheology is a recognized and effective technique for obtaining essential insights into the microstructure of polymer blends and composites, as evidenced by various studies [43,44,45,46,47]. Specifically, the prepared samples’ linear viscoelastic properties were investigated and illustrated in Figure 6, Figure 7, Figure 8, Figure 9, Figure 10, Figure 11 and Figure 12.

Figure 6 shows complex viscosity (ղ*), storage modulus (G′), and loss modulus (G″) versus frequency plots. Based on the data presented in Figure 6, it is evident that the introduction of 0.25 wt.% of GO into the PLA/EVOH blend led to a significant reduction in ղ*, G′, and G″, especially at lower frequencies. Upon elevating the GO content to 0.5 wt.%, the ղ*, G′, and G″ values of the PLA/EVOH blend exhibited an increase compared to both the (PLA/EVOH)/0.25GO and the unmodified PLA/EVOH blend. Furthermore, the increase in the GO content to 1 wt.% resulted in elevated ղ*, G′, and G″ values compared to all other samples’.

**Figure 6 polymers-16-01061-f006:**
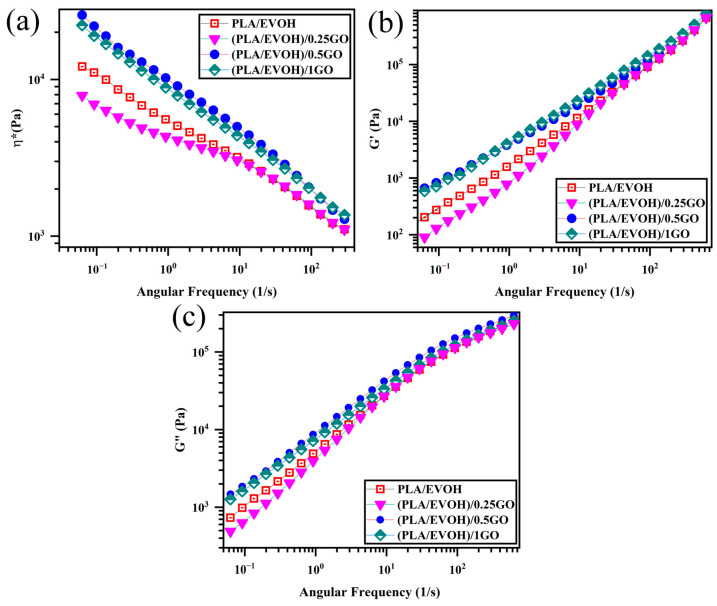
Plots of (**a**) complex viscosity, (**b**) storage modulus, and (**c**) loss modulus versus frequency or PLA/EVOH, (PLA/EVOH)/0.25GO, (PLA/EVOH)/0.5GO, and (PLA/EVOH)/1GO samples.

The decrease in ղ*, G′, and G″ values for (PLA/EVOH)/0.25GO are associated with reduced elasticity. This characteristic can be attributed to the sliding effect (lubricating effect) of the introduced GO [48]. As evidenced by the agglomerates visible in Figure 5, the multilayered structure of stacked GO would slide over each other under shear force when applied to the sample. This sliding process occurred during rheometry and the application of shear to the sample, particularly in layered nanoparticles embedded in the matrix, resulting in reduced viscosity [49,50,51].

Moreover, the dynamics of polymer chains decelerate due to strong interactions between GO and polymer chains (GO/polymer chain interaction) in its presence. It is important to emphasize that while the hydrodynamic interaction of GO and the GO/polymer interaction might enhance elasticity, the lubricating effect of GO is notably suggested to be more effective than other parameters inducing elasticity for (PLA/EVOH)/0.25GO.

The elevation of ղ*, G′, and G″ values in the (PLA/EVOH)/0.5GO samples can be attributed to the predominance of elasticity-inducing factors, such as interactions between GO and polymer chains and hydrodynamic interactions, surpassing the lubricating effect of the GO particles. As mentioned in the morphological section, the increase in the GO content allows the GO particles to migrate toward the interface and within the PLA, facilitating a more widespread distribution of GO within the PLA/EVOH blend. Because GO is present in both the PLA phase and the interface, it can interact with the EVOH, the PLA chains, and the interface. This interaction leads to enhanced elasticity within the matrix and the interface and an increase in droplet elasticity. Due to the rise in the sources inducing elasticity, the increase in elasticity dominated the lubricating effect of GO. Subsequently, the ղ*, G′, and G″ values increased compared to (PLA/EVOH)/0.25GO and the neat PLA/EVOH blends.

Figure 7 displays the damping factor of the PLA/EVOH blend and the PLA/EVOH/GO composites at different contents of GO. In the field of rheological characterization, the point at which tan δ = 1 represents the equilibrium between viscous and elastic behavior, indicating the transition from liquid to solid behavior or vice versa [52,53]. The (PLA/EVOH)/0.25GO sample demonstrated a significant increment in the intensity of tan δ compared to the neat blend. This increase in the damping factor is attributed to the lower elasticity imposed on the system, which can be related to the lubricating effect of GO on polymer chains, which is consistent with the results discussed above. By increasing the content of GO to 0.5 wt.%, the damping graph shifted toward the lower value, which was higher than the tan δ value of (PLA/EVOH)/1GO. This shift is attributed to the superior interactions between polymer chains and GO, which dominate the lubricating effect of the GO sheets in high concentrations of the GO nanoplatelets. Furthermore, the migration of additional GO facilitates a more widespread distribution within the PLA/EVOH sample, enhancing sources for inducing elasticity. As a result, this increased distribution of GO contributes to a reduction in the segmental motion originating from more interaction between GO and the polymer chains (EVOH, PLA, and interface). 

**Figure 7 polymers-16-01061-f007:**
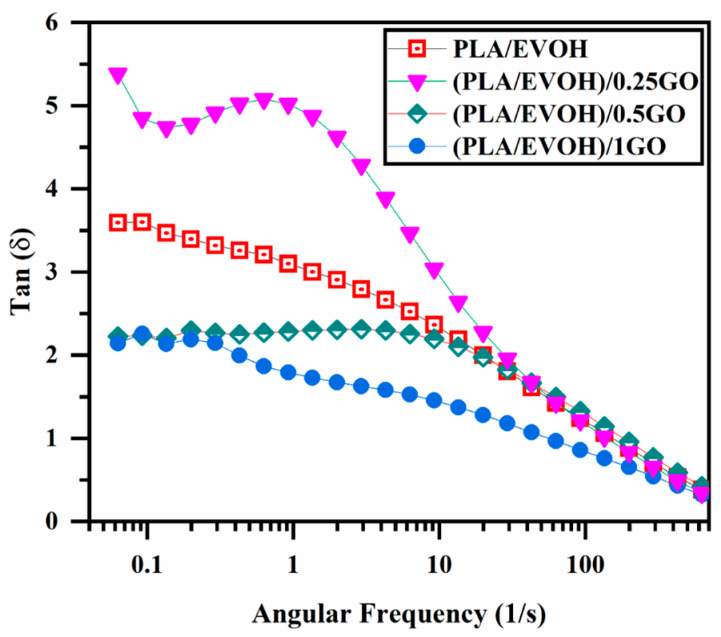
plot of loss tangent (damping factor) versus frequency for PLA/EVOH, (PLA/EVOH)/0.25GO, (PLA/EVOH)/0.5GO, and (PLA/EVOH)/1GO samples.

Figure 8 illustrates the Han plot of the PLA/EVOH blend and its composites. The Han plot, which plots log G′ versus log G″, is a commonly used tool for studying the compatibility and homogenization of multi-phase systems. When analyzing homogeneous samples, such as entirely miscible blends, the terminal region of the Han plot has been observed to have a slope of 2. This characteristic slope provides valuable information on the behavior of the system and can help in optimizing its properties. If the slope of the terminal region in the Han plot deviates from 2, it indicates the presence of heterogeneity in the system. This is commonly observed in immiscible binary polymer blends, where the slope in the terminal region of the Han plot deviates from 2. This deviation clearly indicates the blend’s heterogeneous structure, which can significantly impact its properties and behavior [54,55,56]. When the Han plot exhibits more significant deviation beyond the miscibility criteria, it indicates heightened elasticity in the low-frequency region. It is widely acknowledged that the interface’s elasticity contribution is more significant in the low-frequency range (ω << 1) [57]. 

**Figure 8 polymers-16-01061-f008:**
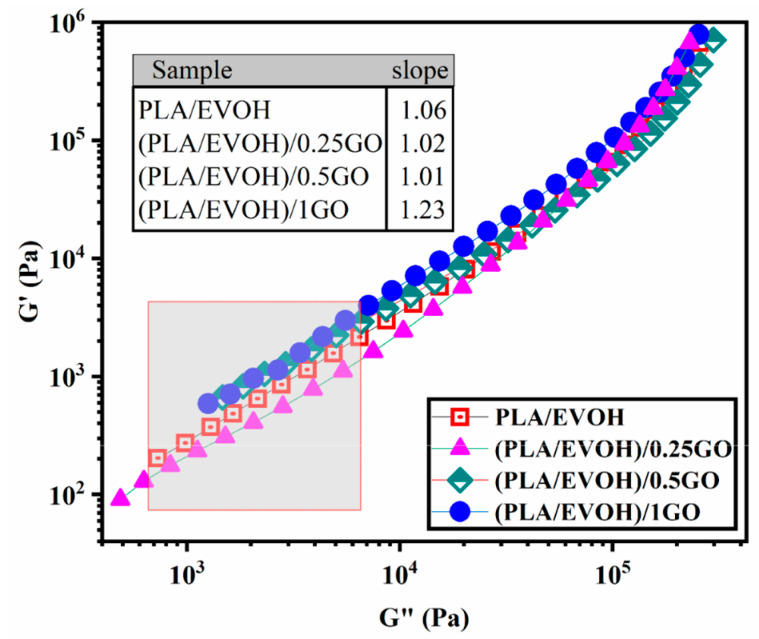
Plots of storage modulus versus loss modulus (Han plots) for PLA/EVOH, (PLA/EVOH)/0.25GO, (PLA/EVOH)/0.5GO, and (PLA/EVOH)/1GO samples.

The slope at low frequencies in the Han plots exhibited values of 1.06, 1.02, 1.01, and 1.23 for PLA/EVOH, (PLA/EVOH)/0.25GO, (PLA/EVOH)/0.5GO, and (PLA/EVOH)/1GO, respectively. The most notable elevation in slope at low frequencies within the Han plots was observed in the sample containing 0.25 wt.% of GO, while the lowest slope was found in the sample with 1 wt.% GO (falling below that of the pure PLA/EVOH blend).

Notably, incorporating GO into the blend led to higher heterogeneity, as evidenced by a more pronounced deviation from the slope of 2 in the rheological analysis. This outcome arises from GO’s impact on the segmental motion within the system. GO’s presence can enhance segmental motion by facilitating the sliding of the GO particles. Yet, this effect can be mitigated through improved GO dispersion due to robust interactions between GO and polymer chains.

Notably, deviations from the slope of 2 were evident in the low-frequency Han plots for all samples, with a more pronounced deviation seen in samples containing GO. Comparatively, the slope of Han plots at low frequencies was increased for (PLA/EVOH)/0.25GO compared to the pure PLA/EVOH blend. However, as the GO content increased to 0.5 wt% and 1 wt%, the slope decreased by 4.7%.

The increase in the initial slope of the Han plot can be attributed to a reduction in elasticity, primarily influenced by the dominant lubricating role of GO, which mitigates the factors contributing to increased elasticity. As the amount of GO increased from 0.25 wt.% to 0.5 wt.%, a significant sharp decrease in the slope of the graph was observed, decreasing from 1.23 to 1.02. In this state, the increase in elasticity can be attributed to the migration of GO toward the interface and PLA, leading to an augmentation of elasticity sources. This increase in elasticity predominates over the chain slippage resulting from GO’s lubricating role.

Figure 9 displays the Cole–Cole plots for all samples, plotting the imaginary viscosity (η″) against the absolute viscosity (η′). The Cole–Cole plot is a well-established technique that provides detailed information on the relaxation processes of polymer blends and filled systems, making it a valuable tool for analyzing the data presented in this study. The Cole–Cole diagram for the neat polymers appears as a circular arc, indicating a homogeneous single-phase structure. In contrast, the neat binary blend and nanocomposite samples showed a deviation from the entire arc, indicative of non-homogeneous systems. This is likely due to additional relaxation processes in these samples [6,58,59].

**Figure 9 polymers-16-01061-f009:**
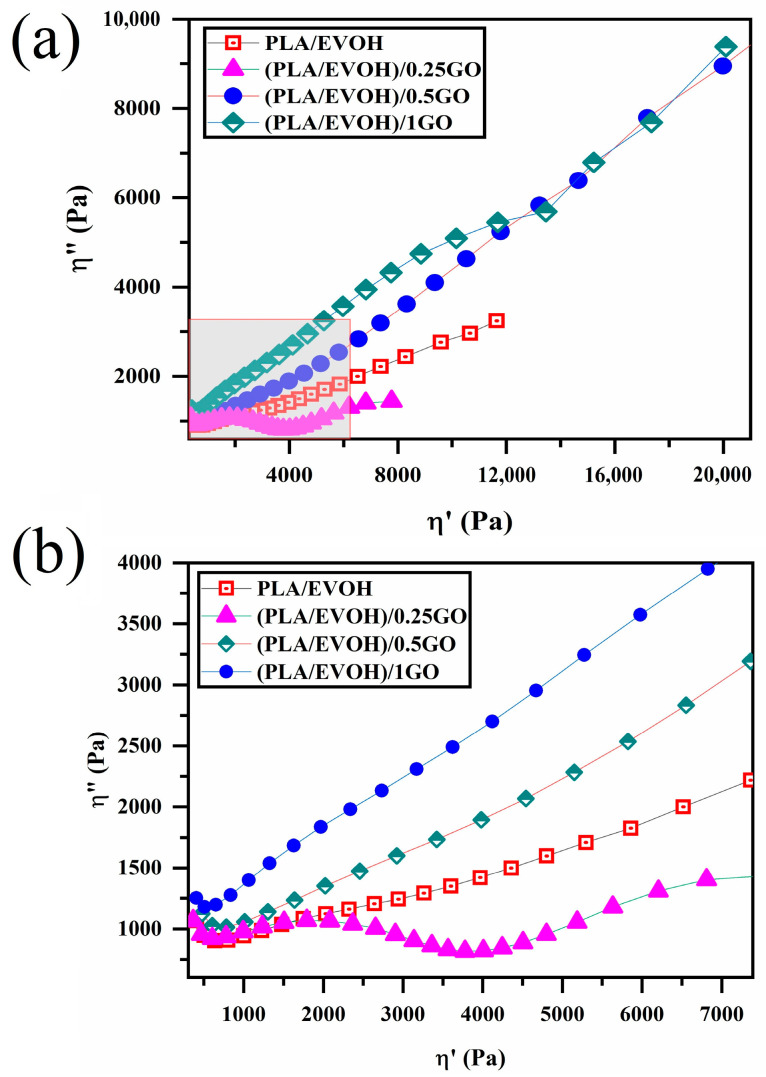
Plots of (**a**) loss viscosity versus storage viscosity (Cole–Cole plots) for PLA/EVOH, (PLA/EVOH)/0.25GO, (PLA/EVOH)/0.5GO, and (PLA/EVOH)/1GO samples and (**b**) magnified view of the rectangular area within (**a**).

Cole–Cole plots can illustrate two semicircles for binary polymer blends. However, in some cases, the semicircle may not be observed. Cole–Cole plots are a valuable criterion for the relaxation analysis of polymer blend composites. The first semicircle is associated with the interaction between the polymer and the particle. The stronger the polymer/particle interaction, the higher the shift of the first semicircle in relaxations. The second semicircle corresponds to droplet relaxation, and the droplet’s size influences its shift to higher relaxations; larger droplets result in a shift to higher relaxations. When nanoparticles are localized at the interface, the tail of the second semicircle is taller [60,61,62,63].

Based on Figure 9, a whole semicircle was not observed in all samples except for the composite containing 0.25 wt.% of GO. A complete semicircle with a tail can also be observed in the case of (PLA/EVOH)/0.25GO.

As it is known, the semicircle portion indicates the elastic response of the material, suggesting that the blend has some degree of structural integrity or ability to recover its shape after deformation [64]. A relatively short tail of (PLA/EVOH)/0.25GO sample extending from the semicircle indicates a viscous component. This implies that the material also exhibits flow-like behavior, associated with energy dissipation and deformation over time. The presence of the tail suggests that the material has a specific viscosity that contributes to its overall rheological behavior. This observation can be ascribed to the lubricating of polymer chains induced by GO, which agrees with the findings mentioned earlier. However, this tail was not observed for other polymer blend composites due to the domination of elastic contribution on the lubricating factor, leading to incomplete semi-arcs. However, this tail cannot be observed in other polymer blend composites, such as (PLA/EVOH)/0.5GO and (PLA/EVOH)/1GO). This is because of the dominance of elastic contributions in the lubrication factor, resulting in an incomplete semicircle for those systems and a sharp and long tail [54,60,61,62].

Figure 10 displays the storage and loss modulus plots versus frequency for the pristine polymer blend and polymer blends with GO. The storage and loss moduli for the polymer blend without GO intersected at a frequency of 135 s^−1^, which indicates a crossover point. At frequencies beyond this point, the storage modulus was more significant than the loss modulus, suggesting predominantly elastic behavior. A lower crossover point indicates that the sample exhibits higher elasticity [65,66,67]. While the addition of (PLA/EVOH)/0.5GO and (PLA/EVOH)/1GO led to a decrease in the crossover point to 92 (1/s), no significant change was observed for (PLA/EVOH)/0.25GO. This can be attributed to the dominance of the lubricating effect of GO over the polymer chain/GO interactions and hydrodynamic interactions in this sample.

**Figure 10 polymers-16-01061-f010:**
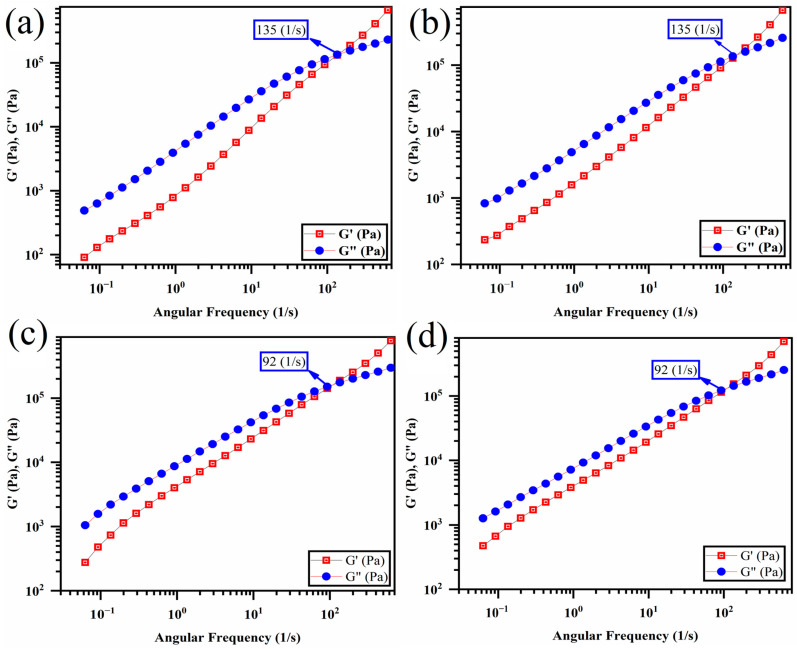
Storage and loss modulus versus frequency intersection (crossover points) for (**a**) PLA/EVOH, (**b**) (PLA/EVOH)/0.25GO, (**c**) (PLA/EVOH)/0.5GO, and (**d**) (PLA/EVOH)/1GO samples.

Figure 11 has been included in this section to examine the impact of mixing protocol on the rheological properties of the PLA/EVOH blends containing GO. Figure 11 displays the complex viscosity, loss factor, Han plots, and Cole–Cole plots of the PLA/EVOH blends containing 0.25 wt.% of GO with different mixing protocols and the neat PLA/EVOH sample.

According to Figure 11a, the (PLA/EVOH)/0.25GO sample exhibited the lowest complex viscosity even than that of the neat blend and (PLA/0.25GO)/EVOH showed the highest viscosity upturn. Referring to Figure 11b, the loss tangent of (PLA/EVOH)/0.25GO was greater than that of the other samples. However, the loss tangent was reduced for both (EVOH/0.25GO)/PLA and (PLA/0.25GO)/EVOH samples. Based on Figure 11c, the slope values of the Han plots were measured as 1.06, 1.23, 1.05, and 1.04 for the PLA/EVOH blend, (PLA/EVOH)/0.25GO, (EVOH/0.25GO)/PLA, and (PLA/0.25GO)/EVOH samples, respectively. And finally, based on Figure 11d, the Cole–Cole plots reveal that (PLA/EVOH)/0.25GO exhibited higher values compared to PLA/EVOH blend, (EVOH/0.25GO)/PLA, and (PLA/0.25GO)/EVOH.

**Figure 11 polymers-16-01061-f011:**
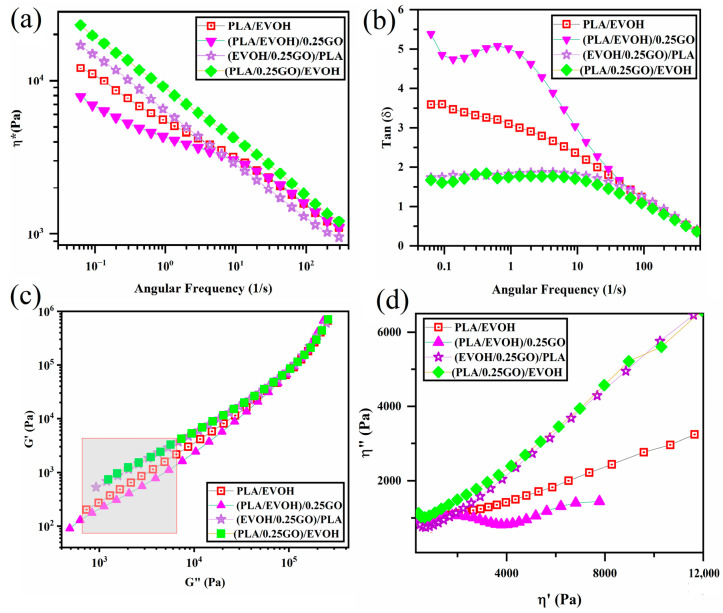
Plots of (**a**) complex viscosity versus frequency, (**b**) loss tangent versus frequency, (**c**) storage modulus versus loss modulus (Han plots), and (**d**) loss viscosity versus storage viscosity (Cole–Cole plots) for PLA/EVOH, (PLA/EVOH)/1GO, (PLA/EVOH)/0.5GO, and (PLA/EVOH)/0.25GO.

According to the findings depicted in Figure 11, (PLA/EVOH)/0.25GO exhibited the lowest complex viscosity, highest loss factor, and reduced elasticity in Han plots, particularly at low frequencies, and the lowest intensity in Cole–Cole plots, especially at high frequencies. The observed decrease in elasticity in the sample (PLA/EVOH)/0.25GO can be attributed to the dominance of the lubricating effect of GO on the PLA/EVOH blend.

On the other hand, when GO was premixed with EVOH ((EVOH/0.25GO)/PLA), GO nanoparticles predominantly localized within the EVOH. As a result, the elasticity of the PLA/EVOH blend increased. This can be attributed to the dominance of hydrodynamic interactions over the lubricating role of GO in this scenario.

When considering the sample (PLA/0.25GO)/EVOH with premixed PLA/GO, the findings indicated the highest level of complex viscosity, the lowest loss factor, reduced elasticity, particularly at low frequencies, and the lowest intensity in Cole–Cole plots. In this mixing protocol, the PLA/GO was prepared first, followed by the incorporation of EVOH. It can be suggested that, although the GO nanoparticles are initially localized within the PLA, as the mixing time progresses, they gradually migrate toward the EVOH and become trapped at the interface layer. Accordingly, the presence of GO at the interface can enhance the adhesion between the two phases and improve interfacial elasticity, which is evident in Figure 11. Additionally, the localization of GO at the interface and PLA prevents the coalescence of droplets, reducing the interfacial tension between the two polymers. This phenomenon is evident in the SEM micrographs, which were discussed earlier.

### 3.5. Rheological Model

To investigate the elasticity of the interface and its contribution to the elasticity of composites, the Lee-Park model was employed, which is derived from the Doi-Ohta model. The Lee-Park model defines G*_d_, G*_m_, and G*_interface_ as the complex moduli of droplets, matrix, and interface, respectively, for polymer blends with G*_b_ of complex modulus and containing a volume fraction φ of the dispersed phase [68,69]. It is noteworthy that the calculation considered the interaction between the GO nanoparticles, as per the works of Strugova et al. [70] Figure 12 shows the contribution of interfacial elasticity as a function of frequency for the PLA/EVOH, (PLA/EVOH)/0.25GO, (PLA/EVOH)/0.5GO, and (PLA/EVOH)/1GO samples. According to Figure 12, there is an increase in the G*_interface_ as the GO content was increased. This observation confirms the earlier concept that a higher GO concentration promotes increased nanoparticle localization at the interface, ultimately leading to more excellent elasticity within the system.

**Figure 12 polymers-16-01061-f012:**
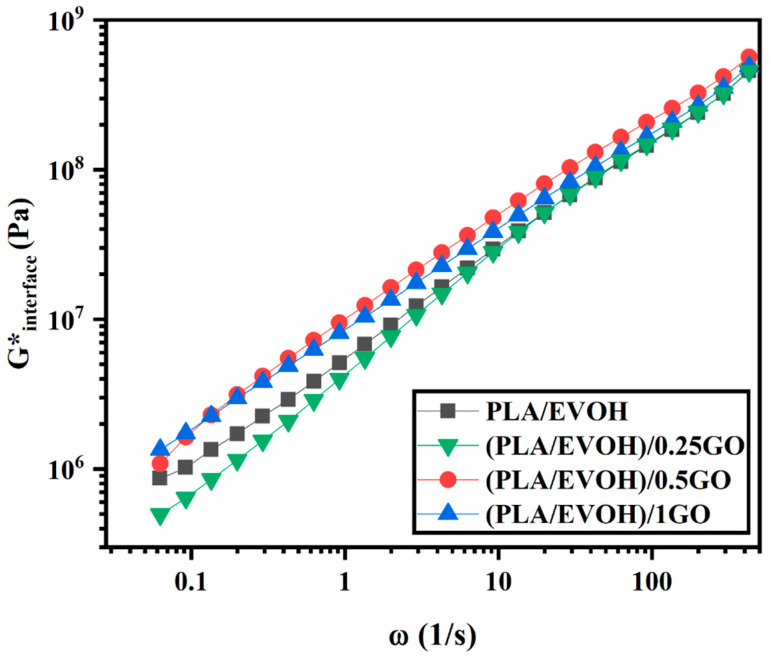
G*_interface_ of Lee–Park model for PLA/EVOH, (PLA/EVOH)/0.25, (PLA/EVOH)/0.5GO, and (PLA/EVOH)/1GO samples.

Figure 13 illustrates the interfacial elasticity of (PLA/EVOH)/0.25GO, (EVOH/0.25GO)/PLA, (PLA/0.25GO)/EVOH, and PLA/EVOH. The results indicate that (PLA/0.25GO)/EVOH, where the GO particles were premixed with PLA, exhibited the highest interfacial elasticity. This can be attributed to the migration of the GO nanoparticles from the PLA phase to the EVOH phase, which, in this way, localized most of the particles at the interface and consequently enhanced the interfacial elasticity.

Figure 14 and Figure 15 are the output of the modified Lee–Park model [71], which illustrate how the elasticity of PLA and EVOH and their blends with the GO nanoparticles correlate. The localization of the GO nanoparticles in either the PLA or EVOH phase can enhance the elasticity of that specific phase.

Aside from the interface between the phases, two primary sources of interaction between GO and the polymers are identified. The Lee–Park model analyses the elasticity of the PLA and EVOH phases, considering their matrix and droplet components. Figure 14 and Figure 15 display contour and 3D plots illustrating the results. Figure 14 and Figure 15 show that increasing the elasticity of only one phase, like PLA or EVOH, does not significantly improve overall elasticity. However, the distribution of GO in each phase can promote the elasticity of that phase. The effect of phase elasticity on total elasticity is suggested for investigation using a modified Lee–Park model. At lower droplet modulus, increasing the matrix modulus does not notably increase elasticity. However, at higher matrix modulus, elasticity becomes more dependent on droplet and matrix elasticity. The same trend is observed for the droplet modulus. The PLA/EVOH blends with more evenly distributed nanoparticles, such as (PLA/EVOH)/0.5GO and (PLA/EVOH)/1GO, exhibit excellent elasticity due to increased elasticity resulting from some GO being localized within the PLA matrix in addition to EVOH. 

### 3.6. Mechanical Properties

Tensile tests were conducted on all samples to investigate the mechanical properties of blends and the impact of the GO content and mixing sequences on the final properties of the blend nanocomposites. Stress-strain plots for all samples are presented in Figure 16. The results are reported in Table 5, providing a comprehensive overview of the data.

The results demonstrate that the elongation at break and tensile strength of the (PLA/EVOH)/0.25GO composites increased, but Young’s modulus is approximately the same compared to the pure PLA/EVOH blend. A notable improvement in the tensile strength, elongation at break, and modulus was observed as the content of GO was increased from 0.5 wt.% to 1 wt.%.

It can be suggested that when the GO nanosheets are dispersed within the polymer matrix, they can act as spacers between polymer chains due to their sliding role. This can increase the separation between polymer chains and decrease the intermolecular interactions contributing to stiffness. As a result, the overall modulus of the blend can decrease. Elongation at the break in the presence of GO increased because the GO sheets can interact with polymer chains very well due to their functional groups. Furthermore, they can suggestively serve as bridges between polymer chains on either side of a crack, absorbing stress and preventing further propagation [72,73,74]. These effects collectively increase the energy required for crack growth, enhancing the material’s elongation at break.

Increasing the GO content to 0.5 wt.% resulted in enhancements in elongation at break and modulus compared to the neat blend and the (PLA/EVOH)/0.25GO composition. This improvement can be attributed to particle localization at the interface and within the PLA phase.

It can be anticipated that by an increase in the GO content, a more uniform and even distribution of nanosheets within the PLA/EVOH blend would be stabilized. Accordingly, this even distribution improves uniform stress transfer throughout the system. Additionally, interfacially localized GO enhances the interfacial adhesion between the two phases, attributed to the amphiphilic and unique chemical structure of GO [27]. This is also evident from morphological observations that revealed a reduction in the EVOH droplet size compared to (PLA/EVOH)/0.5GO. The finer droplet size and uniform distribution contributed to the boosted mechanical properties by facilitating improved stress transfer from the matrix to the droplets and from the GO to the polymer chains. This characteristic makes GO a promising compatibilizer nanoparticle for polymer blends. 

In order to evaluate the impact of mixing on mechanical properties, a comparison was conducted among (PLA/EVOH)/0.25GO, (EVOH/0.25GO)/PLA, and (PLA/0.25GO)/EVOH. Notably, (PLA/EVOH)/0.25GO exhibited superior mechanical properties to (EVOH/0.25GO)/PLA. 

It can be observed that the (PLA/0.25GO)/EVOH sample, in which premixing of PLA and GO was employed, demonstrated the highest mechanical properties. This observation can be attributed to a kinetic effect, which facilitated the localization of GO within the PLA and interfacial regions. Consequently, the adhesion at the PLA/EVOH interface was enhanced, resulting in improved mechanical performance. Moreover, the localization of GO within the PLA phase contributed to an increase in matrix viscosity, subsequently promoting more excellent stress transfer to EVOH and the formation of finer droplets. Consequently, the higher mechanical properties were a direct outcome of more uniform stress transfer from the matrix to the droplet phase, driven by the formation of a finer morphology. Notably, the (PLA/0.25GO)/EVOH sample exhibited the second-highest mechanical properties after (PLA/EVOH)/1GO, which holds the highest rank.

## 4. Conclusions

The PLA/EVOH blends were prepared with a weight ratio of 70/30 (*w*/*w*), where PLA served as the matrix and EVOH as the dispersed phase. The blends were prepared by melt mixing all components simultaneously with different concentrations of GO (0.25 wt.%, 0.5 wt.%, and 1 wt.%). To investigate the effect of the mixing protocol on the rheological, morphological, and mechanical properties, GO was premixed with PLA in one type of mixing and EVOH in another. The wetting coefficient from thermodynamic equations was obtained and equal to 0.01, which showed that GO, thermodynamically, inclines to localize in EVOH. Morphological observations indicated that increasing the GO content to 0.5 wt.% increased the average radius length from 1 to 2.84 µm. This increase was attributed to the partial localization of GO in EVOH. The localization of GO in EVOH was validated by thermodynamic prediction and the TEM micrographs.

The minimum radius length of 0.94 µm was observed for the PLA/GO premixed sample. This decrease was attributed to the predominant interfacial localization resulting from mixing control. On the other hand, the EVOH/GO sample exhibited the highest radius length at 3.13 µm. This sharp increase can be attributed to the predominant localization of GO in the EVOH. Adding 0.5 wt.% and 1 wt.% GO into the blend increased intensity. It shifted the Cole–Cole plots to the right, indicating relaxation at lower frequencies, while 0.25 wt.% GO decreased intensity and shifted the plots to the left, indicating relaxation at higher frequencies. Based on the crossover point analysis of the samples with varying amounts of GO, the addition of 0.5 wt.% and 1 wt.% of GO increased the elasticity.

In comparison, 0.25 wt.% of GO did not significantly affect the crossover point, possibly due to the lubricating effect of GO on hydrodynamic interactions. By premixing GO with PLA before incorporating EVOH, the study found an increase in elasticity due to the dominance of hydrodynamic interactions over the lubricating role of GO. The SEM images showed that the localization of GO at the PLA and interface improved interfacial elasticity but reduced interfacial tension. Incorporating GO into the PLA/EVOH blend increased mechanical properties, including higher Young’s modulus, elongation at break, and tensile strength. Therefore, this exploration is driven by the potential of utilizing these blend composites for packaging applications.

## Figures and Tables

**Figure 1 polymers-16-01061-f001:**
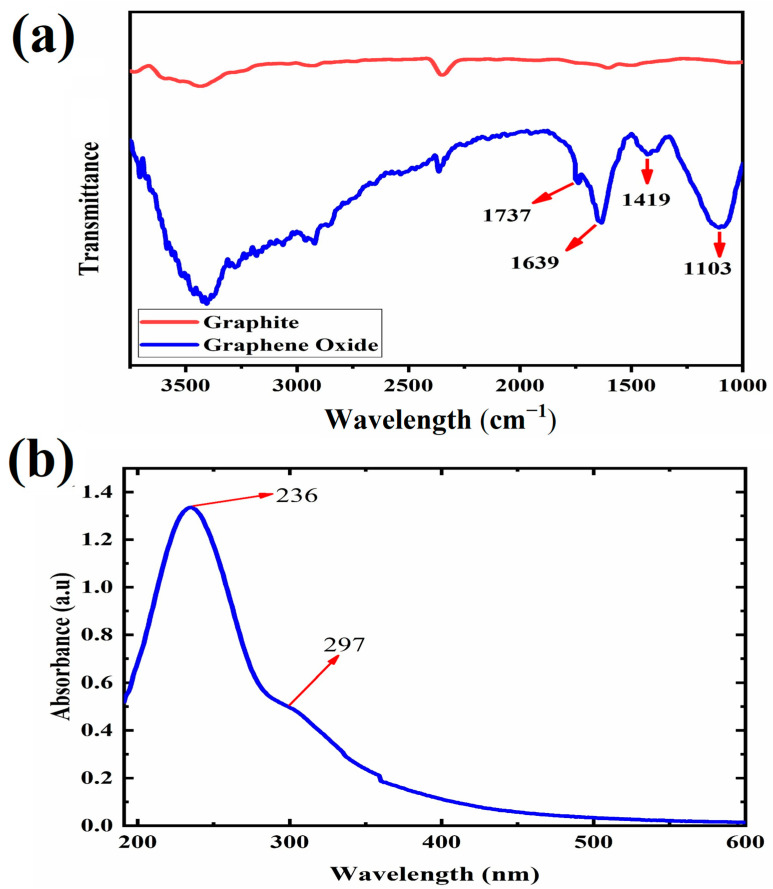
Plots of (**a**) FTIR of graphite, GO, and (**b**) UV-visible characterization of GO.

**Figure 2 polymers-16-01061-f002:**
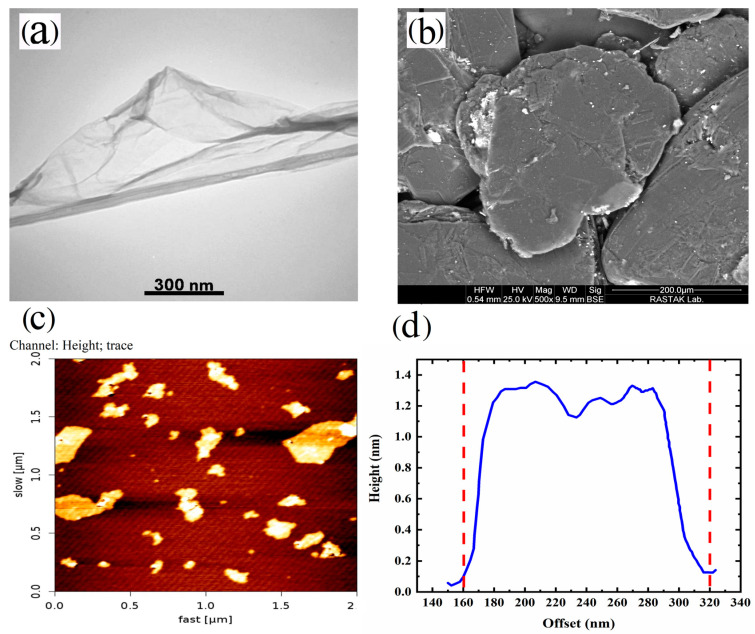
(**a**) TEM micrograph of GO, (**b**) FE-SEM image of graphite, (**c**) AFM image of GO, and (**d**) height profile of GO nanoplatelets.

**Figure 3 polymers-16-01061-f003:**
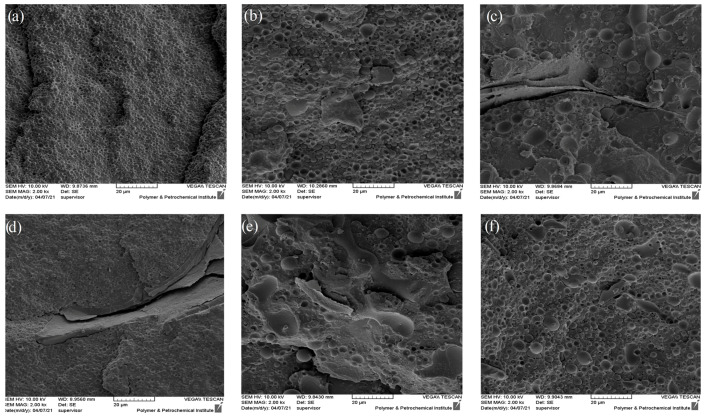
SEM micrographs of the following: (**a**) PLA/EVOH; (**b**) (PLA/EVOH)/0.25GO; (**c**) (EVOH/0.25GO)/PLA; (**d**) (PLA/0.25GO)/EVOH; (**e**) (PLA/EVOH)/0.5GO; (**f**) (PLA/EVOH)/1GO.

**Figure 4 polymers-16-01061-f004:**
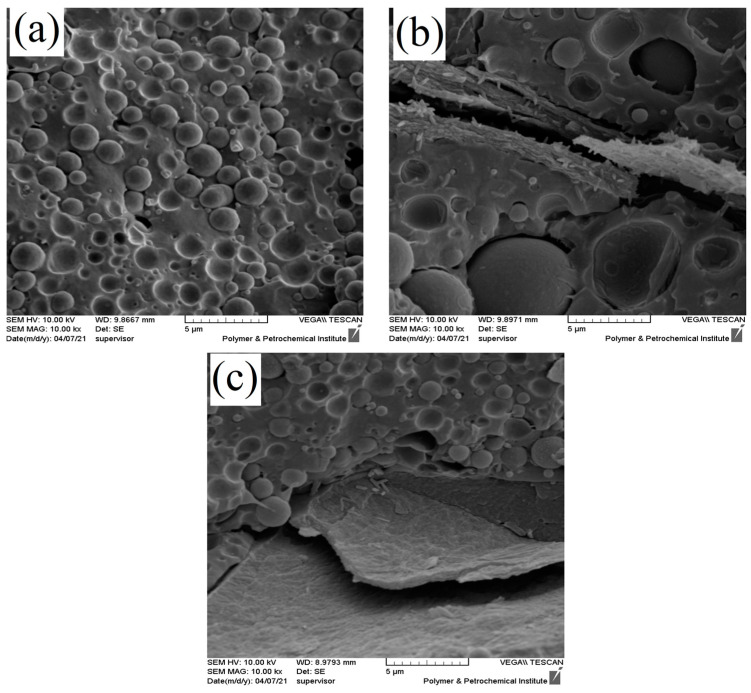
SEM micrographs of (**a**) PLA/EVOH, (**b**) (EVOH/0.25GO)/PLA, and (**c**) (PLA/0.25GO)/EVOH with 10 k magnification.

**Figure 5 polymers-16-01061-f005:**
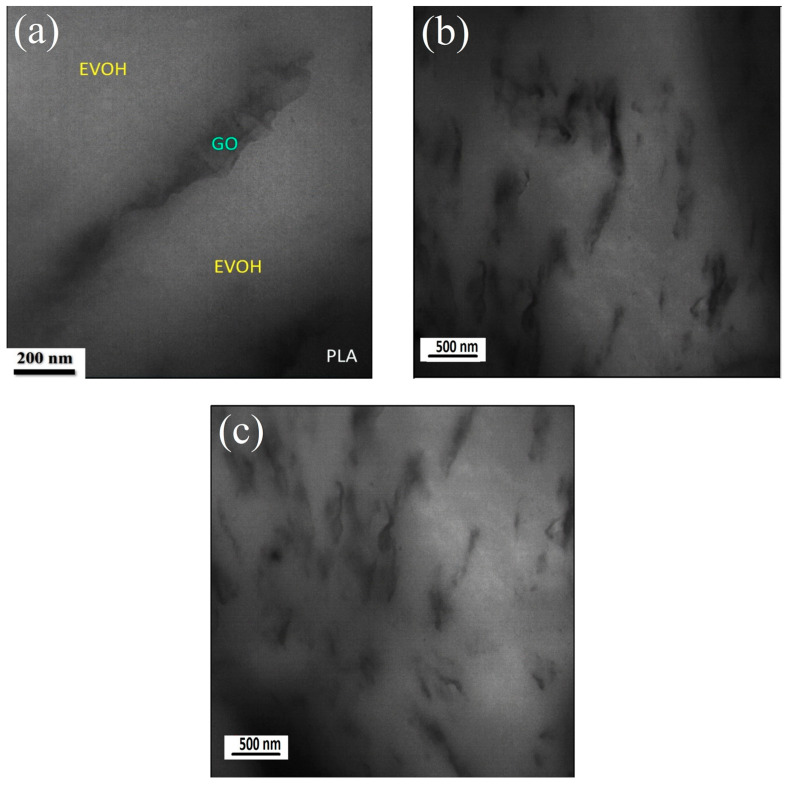
TEM micrograph of (PLA/EVOH)/0.25GO sample, depicting: (**a**) Localization of GO in EVOH; (**b**) Magnification revealing agglomeration in EVOH; (**c**) GO localization in EVOH near the interphase with a lateral size exceeding 10 nm.

**Figure 13 polymers-16-01061-f013:**
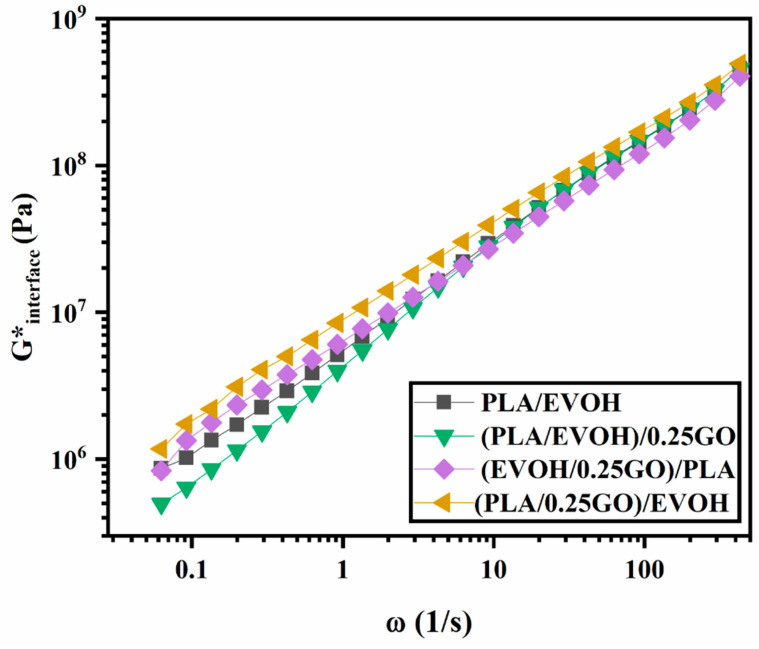
G*_interface_ of Lee–Park model for PLA/EVOH, (PLA/EVOH)/0.25GO, (EVOH/0.25GO)/PLA, and (PLA/0.25GO)/EVOH samples.

**Figure 14 polymers-16-01061-f014:**
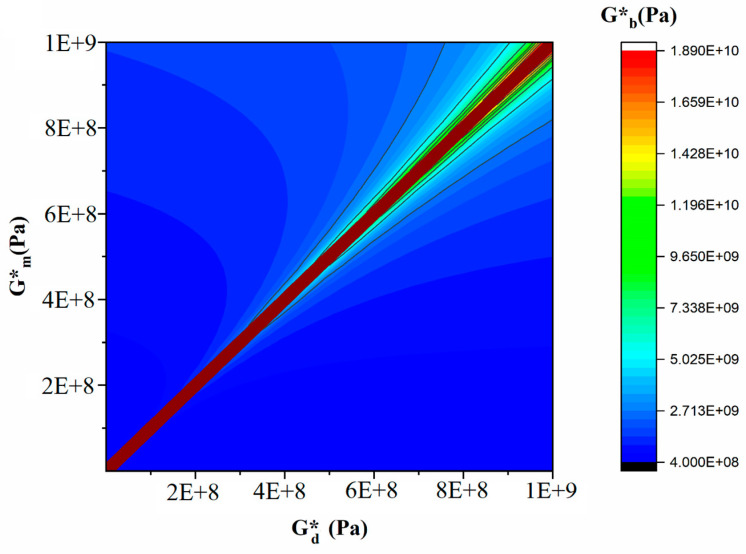
Countorplot of the effect of droplet elasticity and matrix elasticity with polymer blend nanocomposite elasticity.

**Figure 15 polymers-16-01061-f015:**
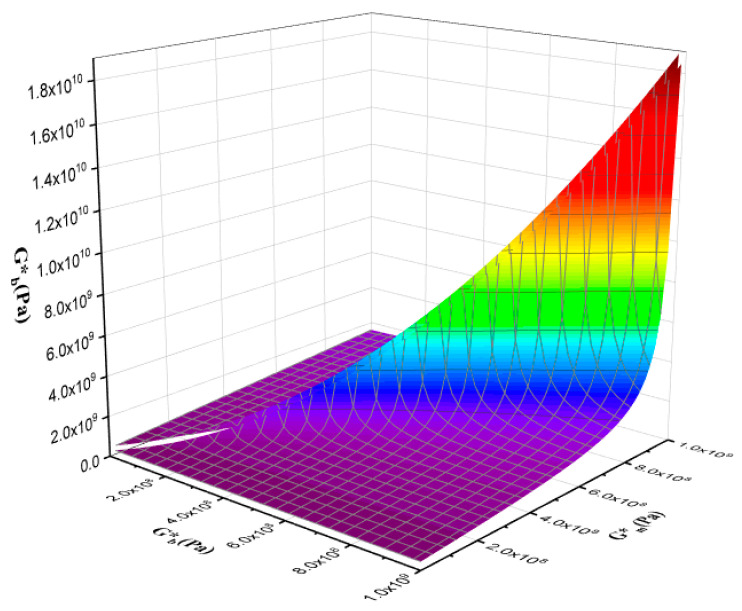
3D plot of the effect of droplet elasticity and matrix elasticity with polymer blend nanocomposite elasticity.

**Figure 16 polymers-16-01061-f016:**
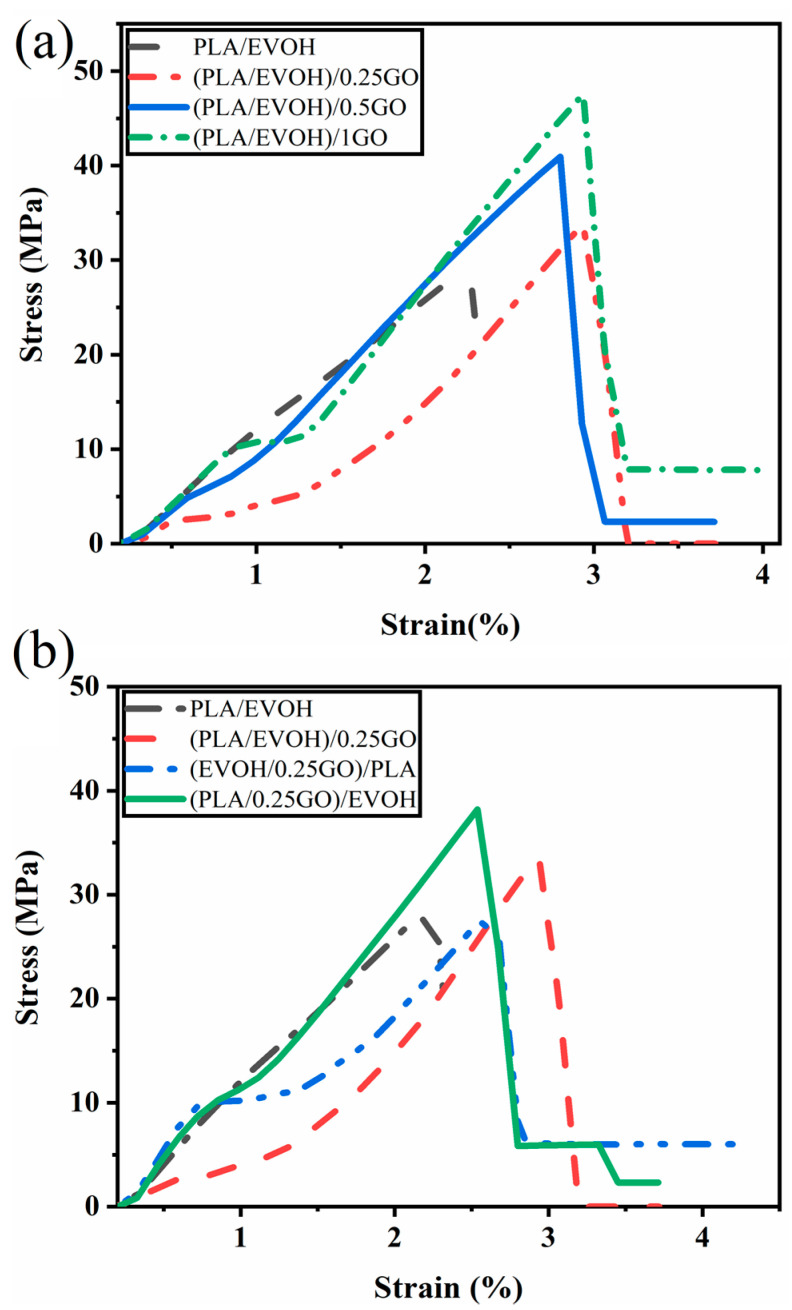
Stress-strain graphs illustrating (**a**) the impact of graphene oxide (GO) content on the mechanical properties of PLA/EVOH composites and (**b**) the influence of mixing protocol on the mechanical properties of PLA/EVOH/GO composites.

**Table 1 polymers-16-01061-t001:** Protocol of mixing, compositions, and code of samples used in this study.

Code	PLA (wt.%)	EVOH (wt%)	GO (wt%)	Protocol of Mixing
PLA/EVOH	70	30	0	M
(PLA/EVOH)/1GO	70	30	1	M
(PLA/EVOH)/0.5GO	70	30	0.5	M
(PLA/EVOH)/0.25GO	70	30	0.25	M
(EVOH/0.25GO)/PLA	70	30	0.25	N
(PLA/0.25GO)/EVOH	70	30	0.25	O

**Table 2 polymers-16-01061-t002:** Surface energies of PLA, EVOH, and GO at 180 °C.

Sample	γ_d_	γ_p_	γ_s_	Reference
PLA	25.1	6.6	31.8	[38]
EVOH	26.4	11.6	38	[39]
GO	32.1	30	62.1	[40,41]

**Table 3 polymers-16-01061-t003:** Interfacial tension of PLA/EVOH, PLA/GO, and EVOH/GO samples.

Sample	Code	Interfacial Tension (Harmonic Equation)	Interfacial Tension (Geometric Equation)
PLA/EVOH	γ_12_	1.5	0.8
PLA/GO	γ_S1_	15.9	8.9
EVOH/GO	γ_S2_	8.7	4.6
	ω_12_	4.2	5.3

**Table 4 polymers-16-01061-t004:** *R*_n_, *R_v_*, and Dispersity of droplets for PLA/EVOH and its nanocomposites.

Sample	*R_n_* (µm)	*R_v_* (µm)	Dispersity
PLA/EVOH	0.81	1.01	1.24
(PLA/EVOH)/1GO	1.26	2.83	2.24
(PLA/EVOH)/0.5GO	1.47	2.84	1.93
(PLA/EVOH)/0.25GO	1.23	1.94	1.57
(EVOH/0.25GO)/PLA	1.60	3.13	1.95
(PLA/0.25GO)/EVOH	0.80	0.94	1.18

**Table 5 polymers-16-01061-t005:** Tensile strength, elongation at break, and modulus for PLA/EVOH blend and PLA/EVOH/GO nanocomposites.

Sample	Tensile Strength (MPa)	Elongation at Break (%)	Modulus (MPa)
PLA/EVOH	27.8 ± 0.1	2.02 ± 0.02	1298 ± 31
(PLA/EVOH)/1GO	44.2 ± 0.05	3.45 ± 0.1	1540 ± 103
(PLA/EVOH)/0.5GO	38.2 ± 0.3	2.94 ± 0.05	1504 ± 98
(PLA/EVOH)/0.25GO	33.8 ± 0.4	3.00 ± 0.01	1293 ± 21
(EVOH/0.25GO)/PLA	27.4 ± 0.4	2.55 ± 0.07	1078 ± 34
(PLA/0.25GO)/EVOH	34.6 ± 0.4	3.47 ± 0.06	1444 ± 78

## Data Availability

Data are contained within the article.

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
