# Peer review of "Effect of Graphene Oxide Localization on Morphology Development and Rheological and Mechanical Properties of Poly(lactic acid)/ethylene vinyl Alcohol Copolymer Blend Composites: A Comprehensive Study"

_polymers, 2024, doi:10.3390/polym16081061_

Round 1

Reviewer 1 Report

Comments and Suggestions for Authors

Review

Manuscript ID: polymers-2938538

Effect of Graphene Oxide Localization on Morphology Development and Rheological and Mechanical Properties of PLA/EVOH Blend Composites: A Comprehensive Study

The manuscript submitted for review is quite long and interesting in places. However, it contains many errors, internal contradictions and repetitions. Below is a far from complete list of comments. I hope that when preparing a new version of the manuscript, the authors will not limit themselves to eliminating only these shortcomings, but will also shorten the manuscript.

Comments

1. When describing the method for obtaining graphene oxide, no references are used. Does this mean that this method was invented by the authors? Let me comment on my question. I have already seen the use of H3PO4 along with H2SO4. I'm not sure that makes much sense. The oxidizing power of a mixture of nitric and sulfuric acids is better than a mixture of H2SO4 and H3PO4. In some studies, the replacement of nitric acid with orthophosphorus was justified by concern for the environment - there is no release of NO2. But washing off phosphates requires a lot of water consumption. In my opinion, the described method is a modification of the Hammers method.

2. I believe that the resulting graphene oxide is not sufficiently qualified. In particular, elemental analysis data (C, H, N, S, P and O) are missing. Without these data, the results obtained can be considered irreproducible.

3. The signature under Table 1 (lines 135-139) repeats lines 126-131. I don't see the point in this repetition.

4. There is no description of sample preparation for recording IR and UV spectra.

5. In the IR spectrum (Figure 1) you can see a wide and intense absorption band in the range of 3500-3250 cm-1, which is due to stretching vibrations of C-H bonds. One might think that the sample under study contains many water molecules. Then, bending vibrations of H2O molecules can contribute to the intensity of the absorption band at 1639 cm-1. Why don't the authors consider this possibility?

6. When assigning peaks in the UV spectrum, the authors provided references to previous works, but when assigning peaks in the IR spectrum there are no references. Why?

7. The caption to Figure 2 says “TEM image of graphite,” but in fact, it shows a SEM image of graphite.

8. Using the AFM method (Figure 2), one GO particle 1.3 nm thick with a lateral size of 160 nm was characterized. Further, these sizes are assigned to all GO particles, although the same figure shows that even in the sample there are particles of different sizes. Moreover, in Figure 4 one can see flat particles with a linear size of 20 μm or more. It is difficult to imagine that the linear dimensions of GO particles increase when they are introduced into a polymer matrix.

9. Lines 181-184. “The presence of darker regions in the TEM micrograph (Figure 2(a)) can be attributed to areas where the thickness of GO is increased due to the presence of wrinkles. These dark regions indicate a higher level of oxidation in the GO resulting from the fabrication process [26]." Here are two explanations for the presence of dark areas in a TEM photograph. I'm willing to believe the first explanation. The second explanation seems dubious and requires clarification.

10. Table 2 reports that “Surface energies of GO at 180 °C” are taken from work [33]. Open work [33]. There is nothing about GO there.

11. Lines 203-204. “when the value of ω falls between -1 and 1, it suggests that the nanoparticles preferentially localize at the interface of polymers.”

Lines 215-217. “ω12 was calculated using equation 1, resulting in a value of 5.3. Therefore, it can be concluded that the GO nanoparticles thermodynamically prefer to localize at the PLA/EVOH interface." What should the reader believe?

12. In my opinion, in formula (5) instead of  one should write  .

13. In my opinion, in the sentence “In this morphology, the polymer with a higher composition (PLA) tends to act as the matrix, while the polymer with a lower composition (EVOH) tends to form the dispersed phase,” the word “composition” is better replace with the word "content".

14. Figure 4 is very interesting. But it would be even more interesting if SEM images of the binary mixtures EVOH/0.25GO, and PLA/0.25GO were presented. By the way, in Figure 4c you can see an GO particle whose thickness is significantly greater than 1.3 nm, and whose literal size is greater than 5 μm. Note that it was previously stated that “based on the observations from Figures 1 and 2, it can be concluded that GO's synthesis successfully produced mainly single-layer platelets.”

15. In my opinion, the spherical particles observed in Figures 3 and 4 cannot be called drops, because a drop is a small volume of liquid. Also a droplet has a spherical shape only in conditions of weightlessness.

16. Once again, I note the internal inconsistency of the manuscript: Lines 263-264 (As evident in TEM results, the thickness of GO nanoplates in the polymer matrix is approximately more significant than 10 nm.) vs Lines 190-191 (it can be concluded that GO's synthesis successfully produced mainly single-layer platelets).

17. It is necessary to describe Õ²*, G’, and G” values at their first appearance in the manuscript (Line 320).

18. The reasoning about the lubricating role of GO seems very long and dubious if one agrees with the statement that “GO's synthesis successfully produced mainly single-layer platelets”

19. One of the samples listed in Table 5 does not contain EVOH by designation.

20. Figure 16 contains horizontal segments, the physical meaning of which is unclear. For some of these segments the “Stress” value is negative!

21. The literature is not formatted according to MDPI rules. Apparently, the manuscript was designed for another publishing house.

22. Ref [60]. It is not clear what the letters (NYNY) mean in the title of the magazine.

Author Response

Reviewer #1:

Effect of Graphene Oxide Localization on Morphology Development and Rheological and Mechanical Properties of PLA/EVOH Blend Composites: A Comprehensive Study

The manuscript submitted for review is quite long and interesting in places. However, it contains many errors, internal contradictions and repetitions. Below is a far from complete list of comments. I hope that when preparing a new version of the manuscript, the authors will not limit themselves to eliminating only these shortcomings, but will also shorten the manuscript.

Response: Thank you for taking the time to review our manuscript. We appreciate your feedback and apologize for any errors, contradictions, or repetitions that may have been present in the initial submission. We will carefully address each of your comments and make necessary revisions to ensure the accuracy and clarity of our work.

In addition to correcting these issues, we acknowledge your suggestion to shorten the manuscript. We aim to present a more concise and focused version of the manuscript in the revised submission.

Comments

  1. When describing the method for obtaining graphene oxide, no references are used. Does this mean that this method was invented by the authors? Let me comment on my question. I have already seen the use of H3PO4along with H2SO4. I'm not sure that makes much sense. The oxidizing power of a mixture of nitric and sulfuric acids is better than a mixture of H2SO4and H3PO4. In some studies, the replacement of nitric acid with orthophosphorus was justified by concern for the environment - there is no release of NO2. But washing off phosphates requires a lot of water consumption. In my opinion, the described method is a modification of the Hammers method.

Response: Thank you for your feedback and for highlighting the importance of referencing the method for obtaining graphene oxide (GO) in our manuscript. We have now included the reference you provided (Marcano et al. methods), which outlines the synthesis method used in our study. Additionally, we appreciate your insights regarding the modification of the Hummers method and the environmental considerations associated with different synthesis routes.

Furthermore, we have acknowledged the conventional nature of this method and its utilization in our previous works, as evidenced by the provided references. We understand the concerns regarding water consumption during the washing off of phosphates and would like to emphasize that we employed filtration methods, including the use of dialysis tubes, to mitigate water usage.

Ultimately, our primary objective in synthesizing GO was to investigate its impact on the mechanical and rheological properties of PLA/EVOH blends. We believe that the chosen method effectively served this purpose, allowing us to achieve graphene oxide with more functionality, as it was mentioned in the literature.

Marcano et al. method:

Daniela C. Marcano, Dmitry V. Kosynkin, Jacob M. Berlin, Alexander Sinitskii, Zhengzong Sun, Alexander Slesarev, Lawrence B. Alemany, Wei Lu, and James M. Tour

ACS Nano 2010 4 (8), 4806-4814

DOI: 10.1021/nn1006368

This conventional method has also been utilized in other works, for instance:

  1. https://doi.org/10.1016/j.surfin.2023.103733
  2. https://doi.org/10.1016/j.susmat.2023.e00755
  3. https://doi.org/10.1016/j.mtchem.2023.101751

2. I believe that the resulting graphene oxide is not sufficiently qualified. In particular, elemental analysis data (C, H, N, S, P and O) are missing. Without these data, the results obtained can be considered irreproducible.

Response: Thank you for bringing this to our attention. We don't believe that this test is essential.

  1. The signature under Table 1 (lines 135-139) repeats lines 126-131. I don't see the point in this repetition.

Response: Thank you for bringing this to our attention. As per your suggestion, we have now removed the repetition of lines 126-131 in the signature under Table 1. We appreciate your careful review of the manuscript and feedback on improving its clarity.

  1. There is no description of sample preparation for recording IR and UV spectra.

Response: We appreciate your keen observation and the opportunity to address this matter. We have now included a description of the sample preparation process for recording IR and UV spectra in the characterization section of the manuscript, as well as highlighted these additions in red for clarity.

FTIR: 2 mg of nanoparticles were mixed with KBr powder and then pressed with hydraulic force to prepare thin circular discs for FTIR analysis.

UV-Vis: 10 mg of GO was dissolved in 100 ml deionized water. The solution was then sonicated for 30 minutes. After sonication, the solution was diluted with deionized water to achieve a proper concentration for UV-vis analysis. Both the solvent and solution were scanned from 190 to 600 nm.

  1. In the IR spectrum (Figure 1) you can see a wide and intense absorption band in the range of 3500-3250 cm-1, which is due to stretching vibrations of C-H bonds. One might think that the sample under study contains many water molecules. Then, bending vibrations of H2O molecules can contribute to the intensity of the absorption band at 1639 cm-1. Why don't the authors consider this possibility?

Response: Thank you for your observation regarding the IR spectrum presented in Figure 1 and for raising the possibility of water molecules contributing to the observed absorption bands. We appreciate your attention to detail and your concern for accuracy in our characterization methods.

To address your concern, we want to clarify our procedures. Before FT-IR analysis, we minimized moisture content by thoroughly drying the mixed powder of graphene oxide (GO) and KBr in a vacuum oven for 24 hours at 30 degrees Celsius. This step was taken to ensure that any water molecules present in the sample were minimized as much as possible.

Upon reviewing the literature and considering your suggestion, we acknowledge that the wide and intense absorption band in the range of 3500-3250 cm-1 could indeed be attributed to O-H bonds rather than C-H bonds.

However, it's important to note that our interpretation of the peaks in the IR spectrum aligns with established literature in the field. We have observed the peak at 1639 cm-1, which is commonly associated with C=O bonds, as reported in the literature. Additionally, our comparison with pure graphite, which did not exhibit a peak at 1639 cm-1, supports our interpretation.

While we acknowledge the possibility of water molecules contributing to the observed bands, we believe that the presence of C=O bonds is a more plausible explanation based on our experimental results and alignment with existing literature.

Thank you for providing references to support your perspective.

References:

  1. 1016/j.apsusc.2015.09.128
  2. 1016/j.ejbas.2016.11.002
  3. 1038/s41598-018-38060-x
  4. 1016/j.pnsc.2015.10.004
  5. 1088/1757-899X/509/1/012119
  6. https://doi.org/10.1007/s42452-019-1188-7
  7. 1166/jnn.2018.14306

  1. When assigning peaks in the UV spectrum, the authors provided references to previous works, but when assigning peaks in the IR spectrum there are no references. Why?

Response: We appreciate your careful observation regarding the references in the assignment of peaks in the IR spectrum. We inadvertently missed including references when assigning peaks in the IR spectrum. Your point is well-taken, and we have added the necessary references to enhance the clarity and credibility of our analysis. The references have been highlighted for easy identification.

 References:

  1. 1016/j.apsusc.2015.09.128
  2. 1016/j.ejbas.2016.11.002
  3. 1038/s41598-018-38060-x
  4. 1016/j.pnsc.2015.10.004
  5. 1088/1757-899X/509/1/012119

  1. The caption to Figure 2 says "TEM image of graphite," but in fact, it shows a SEM image of graphite.

Response: We appreciate your attention to detail in noting the discrepancy in the caption of Figure 2. Indeed, there was an error in labeling the image as a TEM image of graphite when it depicts a SEM image. This mistake has been corrected, and the caption now accurately reflects the SEM nature of the image. 

  1. Using the AFM method (Figure 2), one GO particle 1.3 nm thick with a lateral size of 160 nm was characterized. Further, these sizes are assigned to all GO particles, although the same figure shows that even in the sample there are particles of different sizes. Moreover, in Figure 4 one can see flat particles with a linear size of 20 μm or more. It is difficult to imagine that the linear dimensions of GO particles increase when they are introduced into a polymer matrix.

Response: Thank you for your insightful observation regarding the characterization of graphene oxide (GO) particles using AFM and the interpretation of particle sizes in Figures 2 and 4. We appreciate your attention to detail and dedication to ensuring our research accuracy.

Allow me to clarify the interpretation of the figures:

Firstly, the AFM image in Figure 2 indeed characterizes individual GO particles suspended in water. The thickness of the GO particle measured in this image is 1.3 nm with a lateral size of 160 nm. However, it's important to note that this characterization applies specifically to the individual GO particles observed in the AFM image and not necessarily to all GO particles in the sample.

Secondly, the FE-SEM images presented in Figure 4 are intended to illustrate the morphology of the polymer blends, not the graphene oxide. The flat surfaces observed in Figure 4c are likely related to surface fractures and are not indicative of graphene oxide particles. These SEM images provide valuable information about the morphology of the polymer blends, but they do not directly characterize the graphene oxide particles.

Regarding the variation in particle sizes, it's important to consider the phenomenon of agglomeration when graphene oxide is introduced into a polymer matrix. Due to the higher interaction between GO particles themselves compared to the interaction between GO and the polymer, agglomerates or stacks of GO particles may form. These stacks are demonstrated in Figure 5, where the thickness is approximately 50 to 100 nm, and the lateral size is around 1 micron or slightly larger. This agglomeration phenomenon can lead to variations in particle sizes observed in the sample.

We have added this clarification to the discussion and highlighted it in red to ensure clarity for readers. Thank you again for your valuable input. 

  1. Lines 181-184. "The presence of darker regions in the TEM micrograph (Figure 2(a)) can be attributed to areas where the thickness of GO is increased due to the presence of wrinkles. These dark regions indicate a higher level of oxidation in the GO resulting from the fabrication process [26]." Here are two explanations for the presence of dark areas in a TEM photograph. I'm willing to believe the first explanation. The second explanation seems dubious and requires clarification.

 Response: Thank you for your insightful observation regarding interpreting the dark regions in the TEM micrograph. We acknowledge the validity of both explanations, particularly the presence of wrinkles contributing to darker areas. However, we recognize the potential uncertainty in attributing the darkness solely to oxidation without direct evidence. To ensure clarity, we have eliminated the statement, "These dark regions indicate a higher level of oxidation in the GO resulting from the fabrication process", and focused solely on the explanation related to wrinkles.

Furthermore, we have supplemented our interpretation with additional references to support the notion that darker regions in TEM images can indeed be associated with increased thickness and wrinkling of GO, as commonly observed in the literature. We believe this adjustment enhances the accuracy and clarity of our explanation.

  1. Table 2 reports that "Surface energies of GO at 180 °C" are taken from work [33]. Open work [33]. There is nothing about GO there.

Response: Thanks for your keen review. We have extrapolated GO surface energies from room temperature to 180 C, but 33 references belong to EVOH surface energy, which was inadvertently added to the GO section. We have used the following references for extrapolating surface energy to 180 C. References have been added and highlighted in red.

  1. Lines 203-204. "when the value of ω falls between -1 and 1, it suggests that the nanoparticles preferentially localize at the interface of polymers."

Lines 215-217. "ω12 was calculated using equation 1, resulting in a value of 5.3. Therefore, it can be concluded that the GO nanoparticles thermodynamically prefer to localize at the PLA/EVOH interface." What should the reader believe?

Response: We appreciate your astute observation and the opportunity to address this inconsistency. Upon review, we acknowledge a discrepancy in the interpretation of the wetting coefficient (ω) in the manuscript. The wetting coefficient values calculated using both geometric and harmonic equations were indeed 5.3 and 4.2, respectively, as indicated in Table 3. With a wetting coefficient greater than 1, the thermodynamic preference for the localization of GO nanoparticles is within the EVOH phase as droplets.

We apologize for any confusion resulting from the conflicting statements and assure you that the appropriate changes have been made to ensure accuracy in the revised version of the paper. Correction is highlighted in red.

  1. In my opinion, in formula (5) instead of  one should write .

Response: We are sorry, but we did not understand your comment here.

  1. In my opinion, in the sentence "In this morphology, the polymer with a higher composition (PLA) tends to act as the matrix, while the polymer with a lower composition (EVOH) tends to form the dispersed phase," the word "composition" is better replace with the word "content".

Response: We appreciate your suggestion regarding the use of the term "content" instead of "composition" in the sentence you mentioned. Upon careful consideration, we agree that "content" is a more appropriate term in this context. Therefore, we have made the necessary corrections to the sentence to ensure clarity and accuracy. Your input is highly valued, and we thank you for helping us improve the quality of our manuscript.

  1. Figure 4 is very interesting. But it would be even more interesting if SEM images of the binary mixtures EVOH/0.25GO, and PLA/0.25GO were presented. By the way, in Figure 4c you can see an GO particle whose thickness is significantly greater than 1.3 nm, and whose literal size is greater than 5 μm. Note that it was previously stated that "based on the observations from Figures 1 and 2, it can be concluded that GO's synthesis successfully produced mainly single-layer platelets."

 Response: Thank you for your insightful observation regarding the SEM images in Figure 4 and the interpretation of particle sizes. Your attention to detail is greatly appreciated.

Regarding your comment about Figure 4, while the SEM images provide valuable insights into the morphology of the polymer blends, it's important to clarify that these images are intended to illustrate the overall morphology rather than the localization or characterization of graphene oxide (GO) particles specifically. Therefore, SEM images of binary mixtures such as EVOH/0.25GO and PLA/0.25GO were not included in Figure 4. However, we have included TEM images of (PLA/EVOH)/0.25GO samples to demonstrate the agglomerations, where the lateral size dimensions are about 1 micron and the thickness is about 50 to 100 nm. The flake in Figure 4.c is related to the surface of the broken sample, which we acknowledged when it was sent for FE-SEM microscopy testing.

  1. In my opinion, the spherical particles observed in Figures 3 and 4 cannot be called drops, because a drop is a small volume of liquid. Also a droplet has a spherical shape only in conditions of weightlessness.

Respose: Thank you for sharing your perspective on the terminology used to describe the morphology observed in Figures 3 and 4. We understand your point regarding the definition of a "drop" and its association with a small volume of liquid. However, in the context of polymer blend morphology, the term "droplet" is commonly used to refer to regions of one polymer dispersed spherical particles within another polymer matrix, regardless of their shape or size. Therefore, in our study, the spherical particles observed in Figures 3 and 4 are referred to as "droplets" to describe their dispersed phase within the matrix. We appreciate your attention to detail and value your input.

  1. Once again, I note the internal inconsistency of the manuscript: Lines 263-264 (As evident in TEM results, the thickness of GO nanoplates in the polymer matrix is approximately more significant than 10 nm.) vs Lines 190-191 (it can be concluded that GO's synthesis successfully produced mainly single-layer platelets).

Response: Thank you for your clear explanation regarding the apparent inconsistency in the manuscript. We understand your concern and would like to clarify the context of the statements you highlighted.

The statement "it can be concluded that GO's synthesis successfully produced mainly single-layer platelets" pertains to the characterization of GO nanoparticles in water, as depicted in Figure 2. This characterization is conducted before GO is incorporated into the polymer matrix.

Conversely, the statement "As evident in TEM results, the thickness of GO nanoplates in the polymer matrix is approximately more significant than 10 nm" refers to the observation of GO nanoplates embedded within the PLA/EVOH blend, as shown in Figure 5. In this context, the aggregation of GO nanoparticles within the polymer matrix leads to GO stacks comprising more than 10 layers. This aggregation is attributed to increased interaction between GO nanoparticles compared to the interaction between GO and the polymer matrix.

We appreciate your attention to detail and your efforts to resolve the inconsistency. Your clarification provides valuable context for interpreting the results presented in the manuscript.

  1. It is necessary to describe Õ²*, G', and G" values at their first appearance in the manuscript (Line 320).

Response: Thank you for your suggestion. We have now added the following sentence at the first appearance of complex viscosity (Õ²*), storage modulus (G'), and loss modulus (G") values in the manuscript, as per your recommendation:

"Figure 6 shows complex viscosity (Õ²*), storage modulus (G'), and loss modulus (G") versus frequency plots."

This addition aims to provide clarity and ensure that readers are informed about the significance of these values when they are first introduced. We appreciate your attention to detail and your efforts to enhance the readability of our manuscript.

  1. The reasoning about the lubricating role of GO seems very long and dubious if one agrees with the statement that "GO's synthesis successfully produced mainly single-layer platelets"

Response: Thank you for your feedback regarding the lubricating role of graphene oxide (GO) in the polymer blends. Allow me to address your concerns:

  1. Clarification on GO characterization: The statement about "mainly single-layer platelets" pertains to the characterization of GO in water, not its state after being added to the polymer blend. This distinction is important as GO can undergo agglomeration and layer stacking when incorporated into the polymer matrix, altering its behavior.
  2. Reference to existing literature: Previous studies have indeed investigated the rheological behavior of blends containing graphene or GO, and layered particles, noting a lubricating effect when the graphene is not fully exfoliated. This literature supports our argument regarding the potential lubricating role of GO in our polymer blends.

Here are some new references:

  1. Mendoza-Duarte, M. E., & Vega-Rios, A. (2024). Comprehensive Analysis of Rheological, Mechanical, and Thermal Properties in Poly(lactic acid)/Oxidized Graphite Composites: Exploring the Effect of Heat Treatment on Elastic Modulus. Polymers, 16(3), 431. https://doi.org/10.3390/polym16030431.
  2. de Bomfim, A. S. C., de Oliveira, D. M., Benini, K. C. C. de C., Cioffi, M. O. H., Voorwald, H. J. C., & Rodrigue, D. (2023). Effect of Spent Coffee Grounds on the Crystallinity and Viscoelastic Behavior of Polylactic Acid Composites. Polymers, 15(12), 2719. https://doi.org/10.3390/polym15122719.
  3. Gao, P., Muller, S. E., Chun, J., Zhong, L., & Kennedy, Z. C. (2023). Effects of 2D filler on rheology of additive manufacturing polymers: Simulation and experiment on polyetherketoneketone-mica composites. Polymer, 269, 125722. https://doi.org/10.1016/j.polymer.2023.125722.
  1. Explanation of agglomeration and sliding role: In our polymer blends, when GO particles agglomerate, they can slide on each other due to the stacked layers, potentially leading to a reduction in viscosity. However, it's essential to acknowledge that other factors, such as hydrodynamic interactions and interactions between GO and the polymer, also play a role in determining the final rheological behavior.

By providing this context, we aim to offer a comprehensive understanding of the lubricating effect of GO in our polymer blends. We have incorporated these explanations into the manuscript, highlighting them for clarity.

  1. One of the samples listed in Table 5 does not contain EVOH by designation.

Response: Thank you for bringing this to our attention. We have corrected the error regarding the sample listed in Table 5 that was designated without EVOH. The correction has been made, and the updated information has been highlighted in red to ensure visibility. We appreciate your thorough review of our manuscript, and your diligence in identifying such discrepancies is invaluable in maintaining the accuracy and integrity of our work.

  1. Figure 16 contains horizontal segments, the physical meaning of which is unclear. For some of these segments the "Stress" value is negative!

 Response: Thank you for your keen observation. Yes, you are right; as a consequence of the slip-aging nature of the samples and the slippage during the first tensile test, such curves were obtained. We will reconsider all the data to find better curves. Negative values were related to the apparatus error and have been corrected. However, I should mention that Young's modulus was calculated based on the linear region. The plots have now been considered and represented.

  1. The literature is not formatted according to MDPI rules. Apparently, the manuscript was designed for another publishing house.

Response: Thank you for your response. We declare during submission that we did not format the manuscript according to the MDPI template. We follow this for each of our publications and do necessary journal template-related corrections during revision.

  1. Ref [60]. It is not clear what the letters (NYNY) mean in the title of the magazine.

Response: Thank you for pointing out this in the reference [60]. We have now corrected it by providing a more descriptive title and highlighting the change in red. The revised reference is as follows:

Strugova D, David É, Demarquette NR. Linear viscoelasticity of PP/PS/MWCNT composites with co-continuous morphology. Journal of Rheology. 2022;66(4):671-681.

We appreciate your attention to detail and your efforts to ensure accuracy in our manuscript

Reviewer 2 Report

Comments and Suggestions for Authors

Authors present the influence of GO on the properties and morphology on PLA/EVOH blends.

I have not found any similar article although the combination of PLA and EVOH is not so uncommon. The article is well prepared but sometimes the results are not well explained.

Here are my comments:

Equations in the text are too big compared to other text.

Abstract:

 “The rheology results indicated a decrease in the elasticity for the composite containing 0.25 wt.% of GO compared to the neat blend, which was attributed to the sliding effect of added GO nanoplatelets.”

I find it hard to believe. GO is not flat like graphene, to slide. There are OH, CO, COOH, and epoxy groups which make interactions with the OH groups in EVOH and CO groups in PLA. So, sliding should be explained a bit more, or another explanation should be found.

GO Nanoplatelets Synthesis:

Just an observation and comment, no harm! Interesting GO synthesis. I do not see the reason for sonication, which broke plates. According to my experience sonication is not necessary if GO is not dried. I noticed that this was confirmed in one of the recent articles about the synthesis of GO. However, I do not want to say there is something wrong with it!

L198: According to Young equation 29 (Equation 1), …  What is this 29?

Which sentence is the right one?

L201-203: If the value of ω is greater than 1 or less than -1, it indicates that the nanoparticles tend to localize within the polymer phases. However, when the value of ω falls between -1 and 1, it suggests that the nanoparticles preferentially localize at the interface of polymers.

or

L214-216:  Then,ω was calculated using equation 1, resulting in a value of 5.3. Therefore, it can be concluded that the GO nanoparticles thermodynamically prefer to localize at the PLA/EVOH interface.

I believe that SEM images deserve more explanation. What are these larger spots which are up to 10 and more microns? What are the plates in the cracks in Fig. 3, pictures e and f. Besides, I believe I see larger droplets than calculated.

Figure 5 is not convincing. The magnification is too big. PLA content is 70% and PVOH is distributed in the form of micron-sized droplets. If GO is localised in PVOH droplets then I should see GO in the form of droplets and the space without GO between droplets. (or at least visual differences, such as shown on TOC). I see quite nicely or evenly distributed particles.  In Fig.5a there was supposed to be PLA as a black part. Where is this black part in b and c?

Are Fig 14 and Fig 15 just theoretical plots or practical results? It is stated: “Figures 14 and 15 display the contour plot and 3D plot, respectively, illustrating the results. Interestingly, it was observed that increasing the content of only one phase, such as PLA or EVOH, did not significantly improve the overall elasticity.” However, in the experimental part there is only a 70/30 mixture mentioned, so, no increase of one polymer content. Maybe it is just that the explanation of this is not perfect.

L557: A sentence like this gives me a headache: “The results demonstrate that elongation at break and tensile strength of  (PLA/EVOH)/0.25GO composites increased, but Young modulus decreased from 1298.1 to 1293.4 compared to the pure PLA/EVOH blend.” First, Young modulus determined on a decimal place precision is nonsense. The authors know it as they put normal values in Table 5. Second, the difference is only 0.39%. If 6 parallel measurements had been performed the result could have been reversed. So, it would be more correct to say that the value is approximately the same.

The second decimal place at the tensile strength is also not needed.

Table 5: (EVOH/0.25GO)/EVOH should be (EVOH/0.25GO)/PLA and (PLA/0.25GO)/PLA should be (PLA/0.25GO)/EVOH.

Author Response

Authors present the influence of GO on the properties and morphology on PLA/EVOH blends.I have not found any similar article although the combination of PLA and EVOH is not so uncommon. The article is well prepared but sometimes the results are not well explained.

Response: Thank you for your feedback and comments. We're committed to ensuring that our results are thoroughly explained and understandable. 

Comments

Here are my comments:

  1. Equations in the text are too big compared to other text.

 Response: Thank you for bringing this to our attention. We have adjusted the size of the equations to ensure they are consistent with the surrounding text. We appreciate your feedback.

  1. Abstract:

 "The rheology results indicated a decrease in the elasticity for the composite containing 0.25 wt.% of GO compared to the neat blend, which was attributed to the sliding effect of added GO nanoplatelets."

I find it hard to believe. GO is not flat like graphene, to slide. There are OH, CO, COOH, and epoxy groups which make interactions with the OH groups in EVOH and CO groups in PLA. So, sliding should be explained a bit more, or another explanation should be found.

 Response: Thank you for your insightful input and elaboration on the interactions between graphene oxide (GO) and the polymer matrix. I appreciate your attention to detail and dedication to ensuring our research accuracy.

You rightly pointed out that GO, while not completely flat like graphene, still possesses a flake-like structure that can facilitate sliding between layers. This sliding phenomenon can occur when GO particles agglomerate and form stacked layers, as observed in Figure 5. Despite the presence of functional groups on GO, these interactions may not prevent stacking when the layers are numerous, as seen in our TEM analysis.

Regarding the interactions between GO and the polymer matrix, such as PLA and EVOH, you highlighted the importance of considering the dominance of these interactions over GO-GO interactions. In cases where the functional groups on GO have significant interactions with the polymer matrix, they may indeed influence the stacking behavior and prevent excessive layer formation. However, in our system, the observed stacking of GO layers indicates that these interactions may not fully inhibit the stacking process.

Furthermore, you mentioned a new work (not our team) demonstrating a significant sliding effect between PLA and GO in solution casting is intriguing and adds valuable insight into the behavior of GO in polymer matrices. Additionally, observations of sliding effects despite the presence of functional groups in PA6 and GO further support the notion of sliding behavior in layered nanoparticles.

  1. Mendoza-Duarte, M. E., & Vega-Rios, A. (2024). Comprehensive Analysis of Rheological, Mechanical, and Thermal Properties in Poly(lactic acid)/Oxidized Graphite Composites: Exploring the Effect of Heat Treatment on Elastic Modulus. Polymers, 16(3), 431. https://doi.org/10.3390/polym16030431
  2. Dadashi, P., Babaei, A., & Khoshnood, M. (2023). Investigating the role of PA6/GO interactions on the morphological, rheological, and mechanical properties of PA6/ABS/GO nanocomposites. Polymer-Plastics Technology and Materials, 62(6), 756–770. https://doi.org/10.1080/25740881.2022.2133617

We have incorporated these explanations and references into the manuscript, highlighting them for clarity and ensuring a comprehensive understanding of the sliding phenomenon in our polymer blends. Thank you again for your valuable contribution to our research.

  1. GO Nanoplatelets Synthesis:

Just an observation and comment, no harm! Interesting GO synthesis. I do not see the reason for sonication, which broke plates. According to my experience sonication is not necessary if GO is not dried. I noticed that this was confirmed in one of the recent articles about the synthesis of GO. However, I do not want to say there is something wrong with it!

 Response: Thank you for sharing your observation and comment regarding our graphene oxide (GO) synthesis process. We appreciate your input and perspective on this matter. While sonication may not be necessary for certain GO synthesis methods, we have found it beneficial in ensuring uniform dispersion and exfoliation of graphite flakes in our other upcoming works. We utilize this synthesized GO in various applications, including solution casting with chitosan and biodegradable polymers. Your feedback provides valuable insights that we can consider as we continue our research and experimentation in this area. We welcome the opportunity to learn from your experience and explore different approaches to GO synthesis. Thank you once again for your comment.

  1. L198: According to Young equation 29 (Equation 1), …  What is this 29?

Response: We appreciate your keen eye for detail and your valuable feedback on this matter. The reference to "Equation 29" in line 198 was a writing error, and we apologize for any confusion it may have caused. We have now corrected this mistake.

  1. Which sentence is the right one?

L201-203: If the value of ω is greater than 1 or less than -1, it indicates that the nanoparticles tend to localize within the polymer phases. However, when the value of ω falls between -1 and 1, it suggests that the nanoparticles preferentially localize at the interface of polymers.

or

L214-216:  Then,ω was calculated using equation 1, resulting in a value of 5.3. Therefore, it can be concluded that the GO nanoparticles thermodynamically prefer to localize at the PLA/EVOH interface.

Response: We appreciate your diligence in highlighting this issue. The first sentence is correct. It was a writing mistake on our part. The wetting coefficient (ω), calculated using equation 1, resulted in a value of 5.3, as stated in Table 3. With a wetting coefficient greater than 1, the GO nanoparticles thermodynamically prefer to localize in EVOH. We apologize for any confusion caused by the initial statement in the manuscript. This correction has been properly noted, and the appropriate changes have been made to ensure accuracy in the revised version of the paper.

  1. I believe that SEM images deserve more explanation. What are these larger spots which are up to 10 and more microns? What are the plates in the cracks in Fig. 3, pictures e and f. Besides, I believe I see larger droplets than calculated.

Response: Thank you for your insightful feedback. We have taken your suggestions into consideration and made revisions accordingly. We have added more explanation about the SEM images to provide a clearer understanding of the larger spots and plates observed in Figures 3e and 3f. Additionally, we have rewritten the discussion section and utilized bullet points to clarify the morphological results better.

Regarding the appearance of cracks in the SEM images, it is important to note that microscopy on polymer blend samples requires fracturing them after immersion in liquid nitrogen, followed by quick breakage. This process can result in the appearance of cracks on the sample surface alongside the observed droplets and other morphological features of the polymer blends.

As for the larger droplets observed, we have calculated their size using ImageJ software. We would be happy to provide you with the raw data of the droplet selection, including all calculations, in Excel format for further examination.

We appreciate your attention to detail and your valuable feedback, which has helped us improve the clarity and thoroughness of our manuscript. If you have any additional questions or suggestions, please feel free to let us know. Your input is greatly appreciated.

  1. Figure 5 is not convincing. The magnification is too big. PLA content is 70% and PVOH is distributed in the form of micron-sized droplets. If GO is localized in PVOH droplets then I should see GO in the form of droplets and the space without GO between droplets. (or at least visual differences, such as shown on TOC). I see quite nicely or evenly distributed particles.  In Fig.5a there was supposed to be PLA as a black part. Where is this black part in b and c?

Response: Thank you for your insightful observation regarding Figure 5. We acknowledge your concerns about the magnification of the TEM image and the clarity of the features observed. We primarily utilized TEM to observe the graphene oxide's (GO) thickness when added to polymers, which may explain the higher magnification used.

Regarding the distribution of GO within the PLA/EVOH blend, we understand your expectation of seeing GO localized within the EVOH droplets. While we attempted to capture this phenomenon in Figure 5, we recognize that the resolution may not be sufficient to clearly distinguish between GO localized within EVOH droplets and the surrounding matrix.

As for the absence of the black PLA region in Figures 5b and 5c, we apologize for any confusion. The reference to the dark area (Figure 5b top and right) in the image was intended to indicate the presence of the PLA matrix. Still, we understand that it may not be visible at this magnification. We attempted to achieve a higher magnification for better visualization, but limitations such as accessibility and cost hindered our ability to do so.

However, regarding the localization of GO in EVOH, we have other validations like thermodynamic predictions, morphological observations, and rheological assessments, which are the aim of our work. However, we have revised the explanation for clarity and highlighted it in red. We appreciate your feedback and understanding of the challenges of conducting these experiments. Your insights are valuable, and we will consider them in our future experimental designs

  1. Are Fig 14 and Fig 15 just theoretical plots or practical results? It is stated: "Figures 14 and 15 display the contour plot and 3D plot, respectively, illustrating the results. Interestingly, it was observed that increasing the content of only one phase, such as PLA or EVOH, did not significantly improve the overall elasticity." However, in the experimental part there is only a 70/30 mixture mentioned, so, no increase of one polymer content. Maybe it is just that the explanation of this is not perfect.

Response: Thank you for your valuable comment. We have carefully considered your feedback and made necessary revisions to clarify the nature of Figures 14 and 15. The updated explanation, highlighted in red, clearly indicates that these figures depict theoretical assessments rather than practical results.

These theoretical assessments were conducted to estimate the impact of each phase of the polymer blends on elasticity, particularly in comparison to the elasticity achieved when graphene oxide (GO) is used. We have used the modified Lee-Park model to simulate the complex modulus of blends by considering the effect of GO on improving phase elasticity. When GO is distributed to PLA or EVOH, phase elasticity can be improved. The results show that when the elasticity of one phase is improved, it cannot significantly affect the overall elasticity of the polymer blend nanocomposites. However, when the elasticity of both phases is improved as a consequence of introducing GO to each phase as a parameter, which can improve elasticity, it can affect the total elasticity of the system.

The term "content of phase" is inaccurate in explaining our interpretation of results and has been revised accordingly. Yes, it is a theoretical approach based on phenomenological models that can describe experimental observations and aligns with our observations.

Here is our revised version:

Figures 14 and 15 are the output of the modified Lee-Park model [ref], which illustrate how the elasticity of PLA and EVOH, as well as their blends with GO nanoparticles, correlates. The localization of GO nanoparticles in either the PLA or EVOH phase can enhance the elasticity of that specific phase.

Two main sources of interaction between GO and the polymers are identified, aside from the interface between the phases. The Lee-Park model is used to analyze the elasticity of PLA and EVOH phases, considering them as matrix and droplet components, respectively. Figures 14 and 15 display contour and 3D plots illustrating the results. Figures 14 and 15 show that increasing the elasticity of only one phase, like PLA or EVOH, doesn't significantly improve overall elasticity. However, the distribution of GO in each phase can promote the elasticity of that phase. The effect of phase elasticity on total elasticity is suggested for investigation using a modified Lee-Park model. At lower droplet modulus, increasing the matrix modulus doesn't notably increase elasticity. However, at higher matrix modulus, elasticity becomes more dependent on both droplet and matrix elasticity. The same trend is observed for the droplet modulus. PLA/EVOH blends with more evenly distributed nanoparticles, such as (PLA/EVOH)/0.5GO and (PLA/EVOH)/1GO, exhibit excellent elasticity due to increased elasticity resulting from some GO being localized within the PLA matrix in addition to EVOH.

  1. L557: A sentence like this gives me a headache: "The results demonstrate that elongation at break and tensile strength of  (PLA/EVOH)/0.25GO composites increased, but Young modulus decreased from 1298.1 to 1293.4 compared to the pure PLA/EVOH blend." First, Young modulus determined on a decimal place precision is nonsense. The authors know it as they put normal values in Table 5. Second, the difference is only 0.39%. If 6 parallel measurements had been performed the result could have been reversed. So, it would be more correct to say that the value is approximately the same.

The second decimal place at the tensile strength is also not needed.

Response: Thank you for your attention to detail and for bringing this to our notice. We have revised the sentence accordingly to address your concerns. The phrase now reads: "The results demonstrate that the elongation at break and tensile strength of (PLA/EVOH)/0.25GO composites increased, but Young's modulus is approximately the same compared to the pure PLA/EVOH blend." We have also eliminated the second decimal place in the tensile strength value. These changes have been highlighted in red for clarity.

  1. Table 5: (EVOH/0.25GO)/EVOH should be (EVOH/0.25GO)/PLA and (PLA/0.25GO)/PLA should be (PLA/0.25GO)/EVOH.

Response: We appreciate your keen observation. We have made the necessary corrections to Table 5 as per your suggestion. Specifically, we have adjusted "(EVOH/0.25GO)/EVOH" to "(EVOH/0.25GO)/PLA" and "(PLA/0.25GO)/PLA" to "(PLA/0.25GO)/EVOH". These changes have been highlighted in red for clarity.

Round 2

Reviewer 1 Report

Comments and Suggestions for Authors

Review No. 2

The authors agreed in principle with my remarks. If the editors agree with their answers and corrections, then I will also not object to the publication of the manuscript.

I must also apologize for the unfortunate mistake in Remark 5. Thanks to the authors for being sensitive enough to point out my mistake to me. However, if we think logically, then before the question about the deformation vibrations of water molecules, there was no point in mentioning the stretching vibrations of C-H bonds. In fact, remark 5 should look like this: “5. In the IR spectrum (Figure 1) you can see a wide and intense absorption band in the range of 3500-3250 cm-1, which is due to stretching vibrations of O-H bonds. One might think that the sample under study contains many water molecules. Then, bending vibrations of H2O molecules can contribute to the intensity of the absorption band at 1639 cm-1. Why don't the authors consider this possibility? In the new edition of the manuscript, this issue becomes even more relevant, because the authors described the preparation of the sample for recording the IR spectrum as follows: “2 mg of nanoparticles were mixed with KBr powder and then pressed with hydraulic force to prepare thin circular discs for FTIR analysis. Before FT-IR analysis, mixed powder of graphene oxide (GO) and KBr was dried in a vacuum oven for 24 hours at 30 °C to minimize moisture content.” Here the authors did not indicate the weight of the KBr powder, but it was probably at least 200 mg. KBr powder absorbs a lot of water and it is impossible to remove it from KBr powder at 30 °C. But this is not even worth discussing, because the pressing procedure most likely took place in air, since otherwise is not indicated. In this case, the dry powder could again absorb a certain amount of water molecules from the air. I propose to compare the spectrum of GO (Figure 1) and the spectrum of a tablet of pure KBr powder prepared according to the method described in the manuscript. I think that the main contribution to the intensity of the absorption bands at 3500-3250 cm-1 and 1630-1640 cm-1 in Figure 1 comes from water molecules adsorbed by KBr powder.

The comparison of the IR spectra of graphene oxide and graphite in Figure 1 is not discussed in any way in the text of the manuscript and is meaningless because graphite is a conductor. In transmission geometry, thick graphite particles do not transmit IR radiation. It is better to compare the spectra of GO and KBr in Figure 1.

By the way, in the box “2 mg of nanoparticles were mixed with KBr powder and then pressed with hydraulic force to prepare thin circular discs for FTIR analysis. Before FT-IR analysis, mixed powder of graphene oxide (GO) and KBr was dried in a vacuum oven for 24 hours at 30 °C to minimize moisture content" it would be good to indicate the amount of potassium bromide powder and remove the words "with hydraulic force". It is also unclear why write “measurements in the wavenumber range of 400–4000 cm-1” if in Figure 1 the lower limit is 1000 cm-1.

In addition, one last thing. In Figure 1, the Y-axis is labeled “Transmittance (%)”, but there are no percentage numerical values. In connection with the words said above, the spectrum of graphite in the “Transmittance” mode cannot have a higher transmittance, for example, at 4000 cm-1, than the spectrum of GO.

I made these comments solely to further improve the quality of the manuscript under review.

Author Response

Thank you for your understanding and cooperation. We greatly appreciate your constructive feedback, which has contributed to improving the quality and clarity of our manuscript. Your insights have been invaluable in refining our work, and we are committed to addressing any remaining concerns to ensure the manuscript meets the journal's standards. We look forward to your final assessment and hope the revised manuscript meets your expectations.

-Thank you for your detailed explanation and clarification regarding the humidity content in the FTIR test. Your insights into the potential contribution of water molecules adsorbed by KBr powder are valuable, and your suggestion to compare the spectrum of GO with the spectrum of a tablet of pure KBr powder is insightful. Considering the holiday closure of laboratories in Iran (Nowruz celebration), it may not be feasible to conduct the comparison immediately. However, it could be an important follow-up experiment to confirm the source of the observed absorption bands. We appreciate your attention to detail and commitment to ensuring the accuracy of the experimental procedures.

In the literature, GO is typically characterized by FT-IR and UV analysis. We have predominantly utilized these two methods, and you're correct in pointing out the significance of considering humidity. Regrettably, we currently lack access to the lab facilities. Interestingly, other authors may have overlooked this aspect, as you rightly suggest its potential importance. You are correct about the KBr content. It was around 200 mg, and the content has been added and highlighted in red.

-Thank you for your explanation regarding comparing the IR spectra of graphene oxide (GO) and graphite in Figure 1. I understand that the comparison was made to evaluate the functional groups on GO nanoparticles, as referenced in the literature. Given your explanation, it may be beneficial to add a sentence in the manuscript text to clarify the purpose of this comparison and its relevance to the study. Additionally, I appreciate your consideration of comparing the spectra of GO and KBr, although it may not be feasible due to the holiday closures. This comparison could provide further insights into the spectral features observed in Figure 1. 

-Thank you for your feedback. We have updated the text in the box accordingly. The lower limit of the wavenumber range has been corrected to 1000 cm-1, and the phrase "with hydraulic force" has been removed.

-Thank you very much for bringing that to our attention. The Y-axis label in Figure 1 has been updated to "Transmittance" to reflect the data presented accurately.

-Thank you very much for your valuable feedback. We truly appreciate it, and we will certainly utilize it to enhance the quality of the manuscript under review.

Reviewer 2 Report

Comments and Suggestions for Authors

I am quite happy with the explanations and corrections. I believe that the magnification on TEM was too big, not too small. The size of the TEM area is approx 3.5x3.5 microns, which is close to the size of droplet. Or maybe the slices should be thinner. I noticed that you do not say anything about the sample preparation. This should be added. However, I know it is hard to get a TEM on demand, so I am fine with that.

You increased figures and now some looks strange, too large.

Author Response

-Thank you for your valuable comments. We believe they will undoubtedly help us enhance the quality of our manuscript. Regarding the TEM magnification, you are correct that it may be more appropriate to consider adjustments based on the size of the TEM area and the thickness of the slices. As per your suggestion, we have now included information about the sample preparation method for TEM evaluation in the manuscript. We appreciate your understanding of the challenges of accessing TEM facilities on demand.

-Thank you for your feedback. We apologize for any inconvenience caused by the adjustment of the figures. We will review the sizing of the figures to ensure they are appropriate and visually consistent with the manuscript.